# A multiplex method for rapidly identifying viral protease inhibitors

Seo Jung Hong[1,19], Samuel J Resnick [1,2,3,19], Sho Iketani [4,5,19], Ji Won Cha[6], Benjamin Alexander Albert [6,7], Christopher T Fazekas[6], Ching-Wen Chang [5,8], Hengrui Liu[9], Shlomi Dagan[10], Michael R Abagyan[11], Pavla Fajtová[11], Bruce Culbertson[2,12], Brooklyn Brace [1], Eswar R Reddem[13,14], Farhad Forouhar [15], J Fraser Glickman [10], James M Balkovec[8], Brent R Stockwell [9,15], Lawrence Shapiro[4,13,14], Anthony J O'Donoghue[11], Yosef Sabo[4,5], Joel S Freundlich[16,17], David D Ho[4,5,18] & Alejandro Chavez [1,6 ✉]

## Abstract

With current treatments addressing only a fraction of pathogens and new viral threats constantly evolving, there is a critical need to expand our existing therapeutic arsenal. To speed the rate of discovery and better prepare against future threats, we establish a high-throughput platform capable of screening compounds against 40 diverse viral proteases simultaneously. This multiplex approach is enabled by using cellular biosensors of viral protease activity combined with DNA-barcoding technology, as well as several design innovations that increase assay sensitivity and correct for plate-to-plate variation. Among >100,000 compound-target interactions explored within our initial screen, a series of broad-acting inhibitors against coronavirus proteases were uncovered and validated through orthogonal assays. A medicinal chemistry campaign was performed to improve one of the inhibitor's potency while maintaining its broad activity. This work highlights the power of multiplex screening to efficiently explore chemical space at a fraction of the time and costs of previous approaches.

Key words Antiviral; Biosensors; Drug Screening; High-Throughput; Multiplex
Subject Categories Microbiology, Virology & Host Pathogen Interaction; Pharmacology & Drug Discovery

## Introduction

In late 2019, the critical need for preparedness against viral pathogens became clear when SARS-CoV-2 swept across the globe, affecting all facets of life (Wu et al, 2020; Zhou et al, 2020). Yet, it is only a matter of time before another virus with pandemic potential emerges (Woolhouse et al, 2012; Forni et al, 2022). Hence, it is imperative that we consider ways to accelerate the introduction of novel therapeutics to expand our antiviral defenses.

Small molecule therapeutics have proven themselves to be essential weapons within our antiviral arsenal, but the way in which they are discovered has remained largely unchanged for decades, with teams typically screening large chemical libraries against a single viral target (Blair and Cox, 2016; Adamson et al, 2021; Hughes et al, 2011; Berdigaliyev and Aljofan, 2020; Hinkson et al, 2020) (Fig. 1). Individually screening targets cannot keep pace with the number of existing and emerging pathogens and variants, necessitating new methods of searching the vast space of chemical interactions. One potential way to speed up the rate of discovery is to survey multiple targets within a single screen. By applying a multiplex approach to drug screening, the time, effort, and resources spent on each screen can be dramatically decreased, and information on many targets can be obtained within a single assay. This broad survey of small molecule activity is also particularly valuable in the context of pandemic preparedness, where the exact nature of the future threat is unclear, so identifying lead candidates with activity across a range of targets becomes vital (von Delft et al, 2023).

[1]Department of Pathology and Cell Biology, Columbia University Vagelos College of Physicians and Surgeons, New York, NY 10032, USA. [2]Medical Scientist Training Program, Columbia University Irving Medical Center, New York, NY 10032, USA. [3]Department of Medicine, Columbia University Irving Medical Center, New York, NY 10032, USA. [4]Aaron Diamond AIDS Research Center, Columbia University Vagelos College of Physicians and Surgeons, New York, NY 10032, USA. [5]Division of Infectious Diseases, Department of Medicine, Columbia University Vagelos College of Physicians and Surgeons, New York, NY 10032, USA. [6]Department of Pediatrics, University of California San Diego, La Jolla, CA 92123, USA. [7]Department of Cellular and Molecular Medicine, University of California San Diego, La Jolla, CA 92093, USA. [8]Center for Discovery and Innovation, Hackensack Meridian Health, Nutley, NJ 07110, USA. [9]Department of Biological Sciences, Department of Chemistry, and Department of Pathology and Cell Biology, Columbia University, New York, NY 10032, USA. [10]Fisher Drug Discovery Resource Center, The Rockefeller University, New York, NY 10065, USA. [11]Skaggs School of Pharmacy and Pharmaceutical Sciences, University of California, San Diego, La Jolla, CA 92093, USA. [12]Integrated Program in Cellular, Molecular, and Biomedical Studies, Columbia University Vagelos College of Physicians and Surgeons, New York, NY 10032, USA. [13]Department of Biochemistry and Molecular Biophysics, Columbia University, New York, NY 10032, USA. [14]Zuckerman Mind Brain Behavior Institute, Columbia University, New York, NY 10027, USA. [15]Department of Pathology and Cell Biology and Columbia University Digestive and Liver Disease Research Center, Vagelos College of Physicians and Surgeons, Columbia University Irving Medical Center, New York 10032, USA. [16]Department of Pharmacology, Physiology, and Neuroscience, Rutgers University – New Jersey Medical School, Newark, NJ 07103, USA. [17]Department of Medicine, Center for Emerging and Re-emerging Pathogens, Rutgers University – New Jersey Medical School, Newark, NJ 07103, USA. [18]Department of Microbiology and Immunology, Columbia University Vagelos College of Physicians and Surgeons, New York, NY 10032, USA. [19]These authors contributed equally: Seo Jung Hong, Samuel J Resnick, Sho Iketani. ✉E-mail: chavez2@health.ucsd.edu

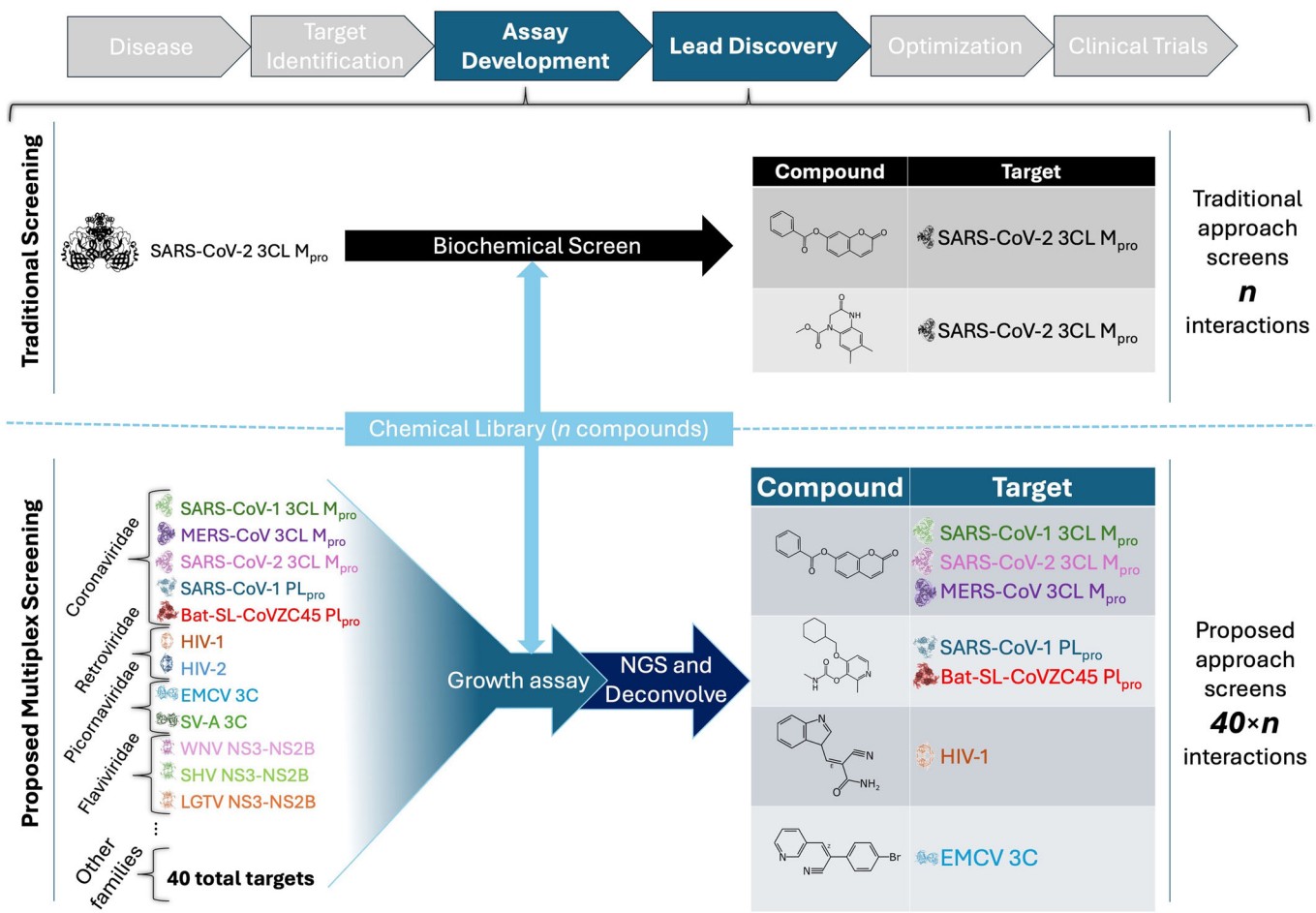

**Figure 1.    Multiplex drug screening platform for viral protease inhibitors.**

A drug screening approach that, unlike traditional screening methods, allows the study of multiple drug targets (i.e., 40 unique viral proteases) simultaneously. In this multiplex screening approach, cells are made to express a drug target of interest (e.g., viral protease), which causes a growth defect that can be rescued upon growing the cells in the presence of a small molecule inhibitor. To enable multiple strains to be tracked at the same time, each cell contains a unique DNA-barcode which allows all the strains to be pooled together and next-generation sequencing (NGS) to be used to quantify the abundance of each member in the pool. Using this approach, we can track interactions between targets and n compounds (e.g., 40 × n).

Here, we present a platform for performing high-throughput multiplex drug screens to identify novel viral protease inhibitors. This platform makes use of DNA-barcoded cellular biosensors to simultaneously screen small molecules against 40 diverse viral proteases (Fig. 1). The use of DNA barcoding enables us to scale our screens an order of magnitude higher than existing antiviral discovery methods that make use of orthogonal fluorescent reporters or secreted analytes (Li et al, 2024; van Rijn et al, 2013; Sarrion-Perdigones et al, 2019). Furthermore, by probing for inhibitors to viral proteases, we have focused our screens on a family of proteins that have a well-established role in the viral lifecycle and are carried by numerous high-priority pathogens (e.g., Crimean-Congo hemorrhagic fever, Zika) (Adamson et al, 2021; Zephyr et al, 2021; Scholte et al, 2017; Russell, 2016; NIAID Emerging Infectious Diseases/Pathogens | NIAID: National Institute of Allergy and Infectious Diseases, 2018).

Our screening approach is based on the phenomenon where the expression of viral proteases within yeast causes a profound growth defect that can be rescued if the protease is inhibited by a small molecule (Blanco et al, 2003; Iketani et al, 2022; Frieman et al, 2011). This cell-based approach allows for the examination of the unmodified viral protease without the arduous task of large-scale protein purification or the need for high-resolution protein structures, as required for in vitro and in silico screens, respectively. In addition, by employing cellular sensors of protease activity, we also eliminate the need for handling live virus; effectively addressing both the biosafety risks associated with many pathogens of interest and the challenges of their cultivation. Additionally, it mitigates unpredictable risks, such as viral recombination between isolates, inherent in a multiplex screening design. To ensure robustness in our multiplex screens, we introduce several design innovations, such as redundant barcoding, the use of multiple in-well controls, and a dynamic hit-calling framework that adjusts for plate-to-plate variation.

Using this novel discovery platform, we tested for compounds with activity against 40 diverse proteases and 3 control non-protease targets. The 40 selected proteases spanned 7 viral families, several of which represent pathogens that have been identified as posing the highest risk to public health (i.e., Category A, B, and C

pathogens according to the National Institute of Allergy and Infectious Disease) (NIAID Emerging Infectious Diseases/ Pathogens | NIAID: National Institute of Allergy and Infectious Diseases, 2018). By screening against a library of 2,480 compounds, we generated 106,640 model-compound interactions and identified numerous putative protease inhibitors. Many of these screening hits showed broad activity against multiple proteases within a given viral family, with some showing effects across families (e.g., coronavirus 3CL and rhinovirus 3C proteases). Due to the proven pandemic potential of coronaviruses, a set of compounds targeting multiple members of the three-chymotrypsin-like (3CL) protease or the papain-like protease (PLP) family were further characterized, and their activity validated via orthogonal biochemical assays (V'kovski et al, 2021; Jin et al, 2020; Shin et al, 2020). Additionally, for the PLP inhibitor, a medicinal chemistry campaign was undertaken to improve its potency and render it functional against authentic SARS-CoV-2. These results demonstrate the power of multiplex small molecule screens in rapidly searching through chemical space to identify broadly active lead compounds that can fill the void in our antiviral therapeutic arsenal.

## Results

### Establishing a multiplex chemical screening platform

Using a drug-sensitized *S. cerevisiae* background (*pdr1Δ pdr3Δ snq2Δ*) (Piotrowski et al, 2017), we developed strains with galactose-inducible expression of either the SARS-CoV 3CL protease, SARS-CoV PLP, or HIV-1 protease. When growing these yeast strains in galactose-containing media, the expression of the unique protease construct within each cell is induced, which consequently reduces their growth due to protease-mediated toxicity. We then verified that incubating these strains with well-defined protease inhibitors (Kim et al, 2012; Blanchard et al, 2004; Ratia et al, 2008) restores growth in a dose-dependent manner (Fig. 2A–D). Having validated yeast as an effective model for detecting protease inhibitors, we proceeded to expand the library to include strains representing 40 unique viral proteases spanning seven different viral families (Fig. 2E). To ensure that the growth inhibition seen upon protease expression was dependent on the catalytic activity of each protease, we generated catalytically-dead mutants for all 40 enzymes. By comparing the growth of the wild-type to the catalytically-dead variant, we confirmed that all the proteases in our library cause toxicity dependent on their proteolytic activity (Appendix Fig. S1), enabling us to use the restoration of cell growth by a compound as a proxy for protease inhibition. As the proteases in this pool caused different levels of toxicity, a series of galactose-inducible promoters of varying strengths were developed. Using these diverse promoters, each protease was tuned to induce a similar level of toxicity so that even highly toxic proteases would not overly deplete and make it difficult to accurately quantify rescue (Appendix Fig. S2).

In addition to our protease-expressing yeast strains, we created a set of control strains. One of these control strains expresses a non-toxic protein, enhanced yellow fluorescent protein (eYFP), to provide a baseline level of growth. The other control strains express "orthogonally toxic" proteins (e.g., YES1 kinase, Appendix Fig. S3) that induce toxicity via mechanisms that differ from the proteases

being studied and can also show rescue by inhibitory compounds. By including these controls, we aimed to eliminate, early in the screening process, compounds that non-specifically rescue yeast growth (e.g., by inhibiting galactose induction or inducing general tolerance to cellular stress). Finally, to survey all the proteases simultaneously, we incorporated DNA-barcoding into our design. In this approach, we "redundantly barcode" each protease or control target-expressing strain by separately transforming them with five stably-replicating plasmids that each contain a unique DNA-barcode. The resulting barcoded strains were then equally combined to generate a final mixed pool of 220 barcoded strains that was used in all subsequent pilot studies and screens. As each strain contains a unique DNA-barcode, we can quantify their proportion in the pool by using targeted amplicon sequencing (Appendix Fig. S4). Furthermore, akin to the use of unique molecular identifiers in CRISPR screens, we can reduce the sources of biological and technical noise by examining the collective behavior of all five DNA-barcoded strains that express the same protease, providing added sensitivity and statistical confidence in our hit calling (Schmierer et al, 2017; Zhu et al, 2019). Before performing a series of unbiased chemical screens, a pilot study was performed in which previously validated protease inhibitors along with the kinase inhibitor, dasatinib (Karaman et al, 2008), were tested against the pool (Fig. 2F). Interactions between our protease models and tested compounds were scored by comparing the barcode abundance of a given protease model in a test well to that of the DMSO control condition within the same screening plate. We then applied a series of normalizations based on the behavior of the various in-well controls built into our pool to prevent spurious hit calling (see Methods for details). A final score for each protease model-compound interaction, which we call a magnitude of effect ratio (magratio), was calculated and used to identify screening hits. Here, a value of 1 indicates inhibition that is comparable to the positive control. Most of the interactions between previously tested protease models and corresponding inhibitors produced magratios near or greater than 1. Additionally, we discovered that these compounds also inhibit related viral proteases (16 additional interactions with magratio scores between 0.22 and 1.84), highlighting the unique ability of the multiplex screening platform to readily detect broad-acting compounds.

### Identification of broad-acting or highly specific protease and kinase inhibitors via multiplex screening

Expanding our screen, we tested 2480 structurally diverse compounds with the goal of identifying inhibitors of the viral proteases in our pool. This set included 1520 compounds from the NCI Diversity Set VI and 960 compounds from the Chembridge DIVERSet-EXP library. The resulting analysis comprised 106,640 total model-compound interactions, and upon ranking by magratios, those representing the top 79 (magratio >0.2), were selected for further validation (Fig. 3A; Appendix Fig. S5). Among these hits were a number of small molecules (CB6728297, CB6762077, CB6778425, NSC138389, NSC287495, and NSC403374) that exhibited broad activity across a range of coronavirus proteases (3CL protease and PLP) and several compounds that appeared to rescue individual models across the viral families: caliciviridae, coronaviridae, flaviviridae, picornaviridae, and retroviridae. Additionally, some compounds selectively rescued only the orthogonally

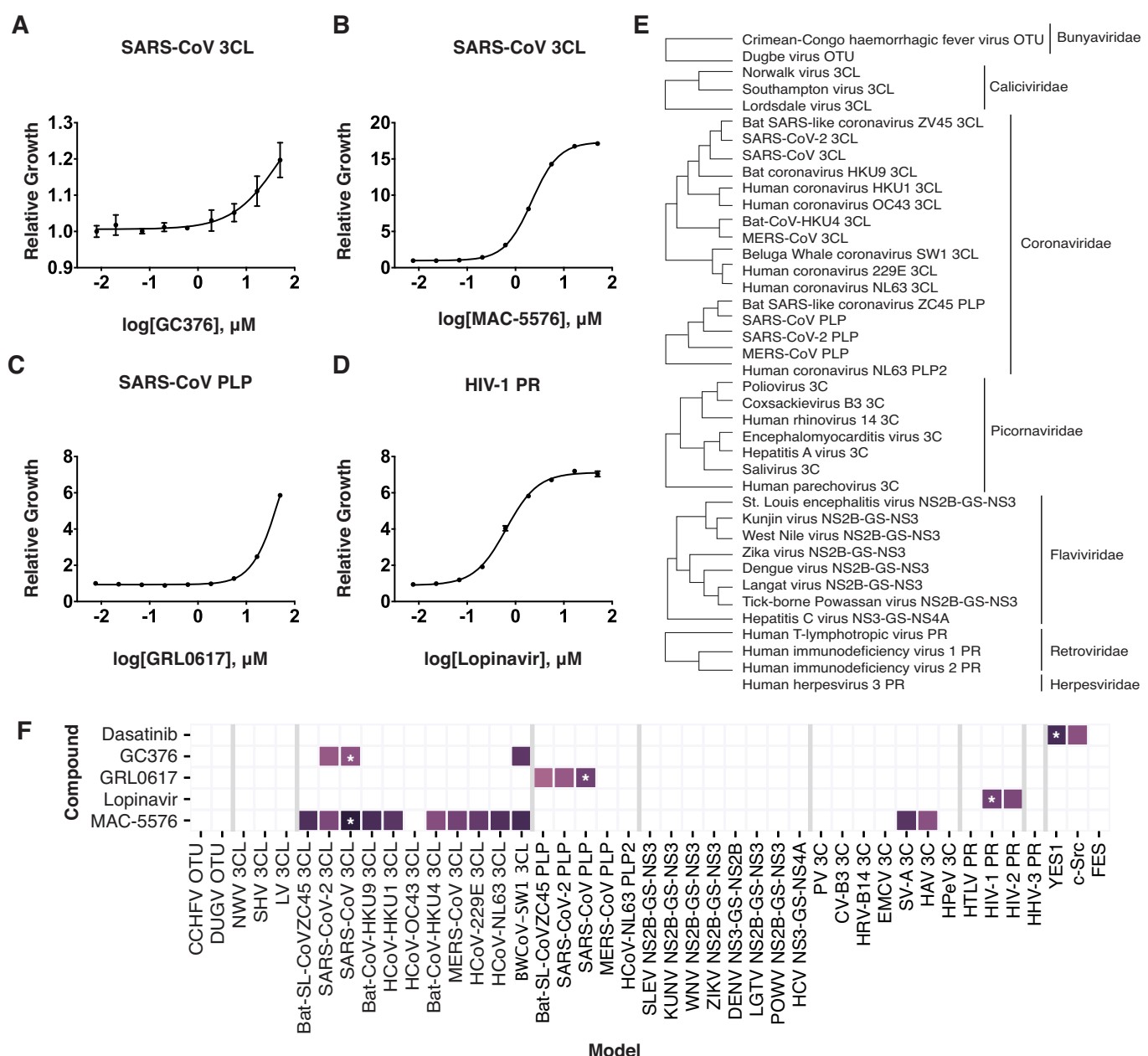

**Figure 2.  Yeast as biosensors for viral protease activity.**

(**A–D**) Dose-dependent rescue of growth in *S. cerevisiae* expressing SARS-CoV 3-chymotrypsin-like (3CL) protease (**A**, **B**), SARS-CoV papain-like protease (PLP) (**C**), and HIV-1 protease (PR) (**D**) by corresponding specific inhibitors. Error bars denote the mean ± s.d. of three technical replicates. Growth curves were determined by nonlinear regression, and the x-axis is in the $\log_{10}$ scale. (**E**) 40 viral proteases represented in the screening pool. OTU, ovarian tumor domain; MERS Middle East respiratory syndrome. (**F**) Migration of model-compound interactions between known viral protease inhibitors and the screening pool. Asterisk (*) denotes expected protease-inhibitor interactions as validated in (**A–D**). CCHFV Crimean-Congo hemorrhagic fever virus, DUGV Dugbe virus, NWV Norwalk virus, SHV Southampton virus, LV Lordsdale virus, SL SARS-like, HCoV human coronavirus, BW Beluga whale, SLEV St. Louis encephalitis virus, KUNV Kunjin virus, WNV West Nile virus, ZIKV Zika virus, DENV Dengue virus, LGTV Langat virus, POWV tick-borne Powassan virus, PV poliovirus, CV Coxsackievirus, HRV human rhinovirus, EMCV encephalomyocarditis virus, SV-A Salivirus, HAV hepatitis A virus, HPeV human parechovirus, HTLV human T-lymphotropic virus, HHV human herpesvirus. Source data are available online for this figure.

toxic kinase control models, suggesting that the platform can be used for detecting drug targets beyond viral proteases.

Upon testing each model-compound interaction within a dose-response assay using the individual yeast strains that harbor the corresponding viral protease or kinase, we were able to determine that the precision of our screen (i.e., true positive/(true positive + false positive)) for all interactions with magratio score >0.2 was 0.61. Furthermore, by increasing the magratio cutoff to 0.3 or 0.4, a corresponding increase in the precision to 0.80 or 0.92, respectively, was observed (Fig. 3B,C; Appendix Figs. S6, S7). Based on these

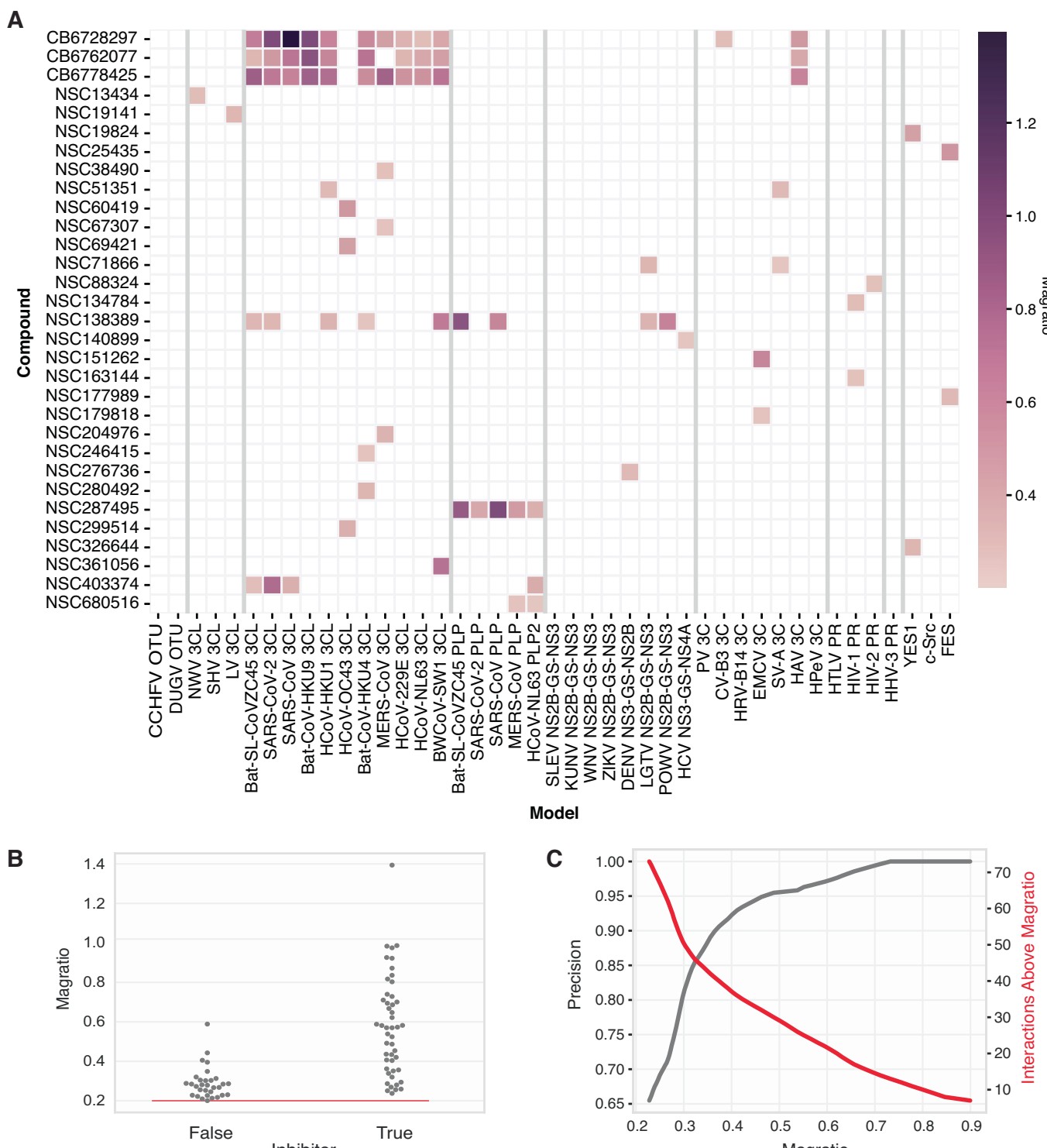

**Figure 3. Screening structurally diverse chemical libraries for viral protease inhibitors.**

(**A**) Top 79 interactions ranked by magratios from screening 2480 structurally diverse compounds from the NCI Diversity set VI and Chembridge DIVERSet-EXP library for small molecules that rescue the growth of models in the screening pool. Interactions shown have magratios > 0.2. (**B**) Compounds from the top 79 interactions were identified as true or false inhibitors via validation in individual target yeast models. (**C**) Precision curve over magratio threshold for tested top interactions (rolling average window of 13). Source data are available online for this figure.

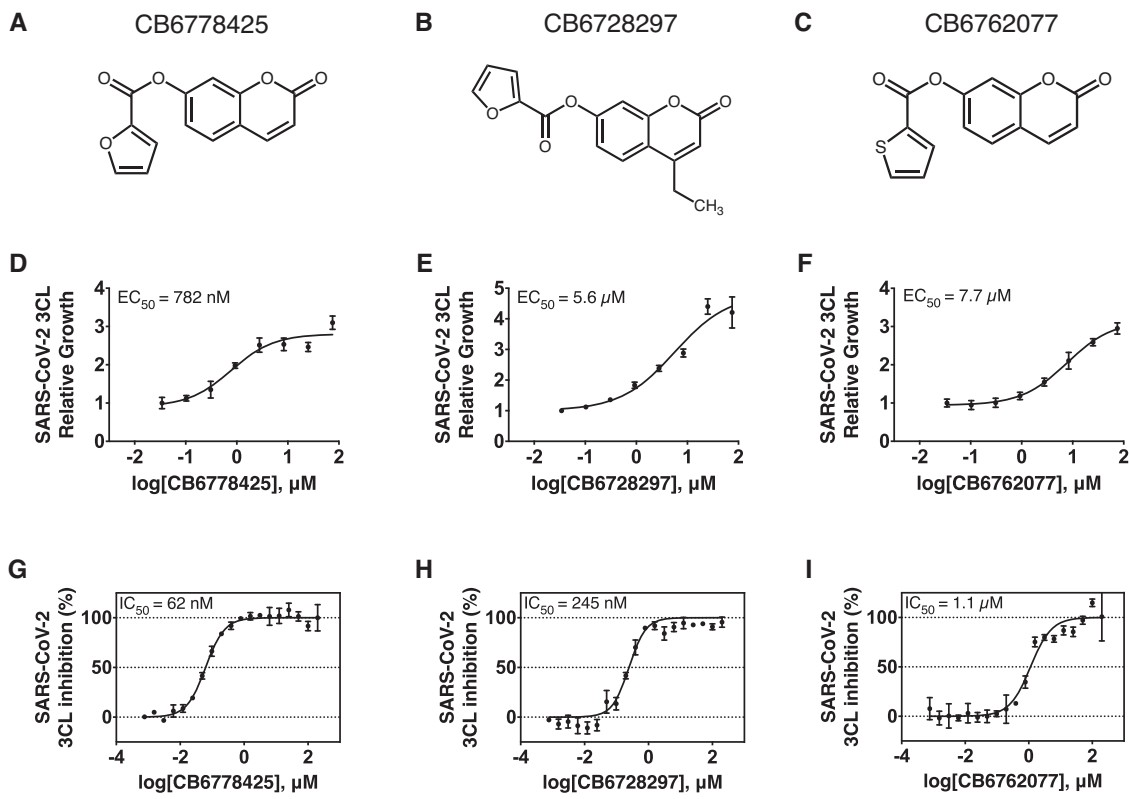

**Figure 4. Orthogonal validation of chromen-2-one-containing inhibitors against SARS-CoV-2 3CL protease.**

(A–C) Chemical structures of CB6778425, CB6728297, and CB6762077. (D–F) Dose-dependent rescue of the SARS-CoV-2 3CL yeast model by the chromen-2-one-containing inhibitors. An assay was conducted with all individually barcoded strains. Error bars denote the mean ± s.d. of five biological replicates. $EC_{50}$ was determined from the growth curve derived from nonlinear regression. The x-axis is in the $log_{10}$ scale. (G–I) Inhibition of purified SARS-CoV-2 3CL protease by the chromen-2-one-containing inhibitors. Error bars denote the mean ± s.d. of two technical replicates. $IC_{50}$ was determined via nonlinear regression. The x-axis is in $log_{10}$ scale. Source data are available online for this figure.

results, in cases where a comprehensive understanding of model-compound interactions is desired, a lower magratio threshold can be set, understanding that this, in turn, will lead to additional false positives. In contrast, if users only desire to identify the most potent interactions, a stricter cutoff can be applied, which will greatly streamline the subsequent validation process at the cost of missing potentially weaker interactions.

## Chromen-2-one family of broad-acting coronavirus 3CL protease inhibitors

The recent COVID-19 pandemic has revealed an ever-present need for preparedness against future outbreaks. Given the continued threat posed by coronaviruses, we focused our efforts on investigating the hits targeting coronavirus proteases, with the goal of exploring their potential as chemical scaffolds for further development. Three chromen-2-one-containing compounds — CB6728297, CB6762077, and CB6778425—improved the growth of a series of coronavirus 3CL protease models as well as two picornavirus 3C protease models in our pool (Coxsackievirus B3 or CV-B3, and Hepatitis A or HAV 3 C proteases) (Figs. 3A, 4A–C). Of note, 3C proteases are structurally related to 3CL proteases and contain a similar overall fold, although they lack a dimerization domain that is present in the 3CL protease family (Yi et al, 2021).

Subsequent validation studies, leveraging the synthesis of these compounds via a two-step route from resorcinol or a one-step route from 7-hydroxycoumarin (Appendix Fig. S8), confirmed the accuracy of the screening results, with the amount of growth rescue showing general concordance with the magratio obtained from the screen (Fig. 4D–F; Appendix Fig. S6A–C). For secondary validation of the target specificity of these chromen-2-one-containing compounds, we purified the SARS-CoV-2 3CL protease and determined the $IC_{50}$ value of each compound against it using a biochemical fluorogenic assay. Dose-dependent inhibition was observed for all three compounds, and the $IC_{50}$ values followed the same trend as the $EC_{50}$ values seen within the yeast system (Fig. 4G–I). The most potent compound, CB6778425, had $IC_{50}$ and $EC_{50}$ values of 62 and 782 nM, respectively, while the least potent compound, CB6762077, had $IC_{50}$ and $EC_{50}$ values of 1.1 and 7.7 μM, respectively. Moreover, a downward shift in the $IC_{50}$s when any of the three compounds were preincubated with the protease suggested that they might function via covalent inhibition (Appendix Fig. S9). To further interrogate this, we conducted mass spectrometry analysis of the SARS-CoV-2 3CL protease treated with CB6778425 and confirmed a ~95 Da shift in the protease-inhibitor complex, suggesting a covalent modification of the protease which we hypothesize to occur between the catalytic Cys145 and the 2-furoyl group of CB6778425 (Appendix Fig. S10).

Initial testing of CB6778425 showed minimal activity within a live virus setting, presumably due to a lability in the reactive 2-furyl ester warhead. Future efforts will be required to determine if this reactivity can be modulated while retaining potent on-target activity.

## Broad-acting coronavirus papain-like protease (PLP) inhibitors

Another notable hit from our screen was NSC287495, a pyridin-4(1H)-one-containing compound that appeared to target a series of coronavirus papain-like proteases (PLPs) (Fig. 3A). Given that there are no clinically approved coronavirus PLP inhibitors, PLP's essential role in efficient viral replication, and the broad activity of NSC287495, we decided to further investigate its therapeutic potential. NSC287495 was synthesized in three steps from the commercially available 3-hydroxy-2-methyl-4H-pyran-4-one via the formation of the corresponding 1-benzyl-3-hydroxy-2-methyl-pyridin-4(1H)-one, installation of the methylcarbamate, and then deprotection of the N-benzyl moiety (Appendix Fig. S11). Validation studies of NSC287495 against each of the coronavirus PLPs in our library verified it to be a broad-acting inhibitor of this family, although the magnitude of inhibition differed across models in line with the screening results (Appendix Fig. S6G).

To identify more potent derivatives, we explored the evolution of the NSC287495 hit. Each synthesized analog was tested against the SARS-CoV PLP and Bat SL-CoVZC45 PLP models, as these showed a robust response to NSC287495 ($EC_{50}$ = 22.8 and 15.6 μM versus SARS-CoV PLP and Bat SL-CoVZC45 PLP, respectively, Appendix Fig. S12). In addition, by testing each derivative against the two PLP models, we aimed to ensure that the breadth of activity was not sacrificed for potency. From our synthesized analogs, a cohort of more potent variants with $EC_{50}$ values in the 0.4–1.7 μM range were uncovered (MAVDA-B-116, -190, -201, -217, and -219) using SARS-CoV and Bat SL-CoVZC45 PLP-expressing yeast models (Fig. 5A,B; Appendix Fig. S12). These hits are members of the 4-alkoxy-2-methylpyridin-3-yl methylcarbamate family (Appendix Fig. S13) and were prepared via variations of the synthetic route to NSC287495 (see Appendix Supplementary Methods for details). All members contain a methylcarbamate which appears to be essential for activity; the corresponding amide and urea analogs were inactive in the yeast assay for SARS-CoV PLP and Bat SL-CoVZC45 PLP (e.g., MAVDA-B-191, 194, and 195; Appendix Fig. S14). We, therefore, hypothesize that a covalent adduct is being formed between these carbamate inhibitors and the protease.

To verify that the restoration in yeast growth was due to direct protease inhibition, in vitro inhibition assays were performed against the purified SARS-CoV-2 PLP enzyme. Similar to the results observed in yeast, all tested compounds were found to inhibit the purified protease ($IC_{50}$ values between 3.5–14.2 μM), with MAVDA-B-219 and B-217 found to be the most potent compounds with $IC_{50}$ values of 3.5 μM and 5.8 μM, respectively. (Fig. 5C). Furthermore, a decrease in $IC_{50}$ values as a function of pre-incubation time with the enzyme suggested a covalent mechanism of inhibition (Appendix Fig. S15A), which was also supported by the observation that the preincubated compound-enzyme complexes, when diluted 100-fold and added to substrate, still exhibited high levels of inhibition unlike GRL0617, a known

non-covalent inhibitor (Appendix Fig. S15B). Finally, we evaluated the performance of the most potent NSC287495 derivatives against authentic SARS-CoV-2 virus. As there can be drastic differences in small molecule retention and behavior in mammalian cell lines, two diverse cell lines, A549 lung adenocarcinoma cells made to overexpress ACE2 and TMPRSS2, and Huh7 hepatocellular carcinoma cells overexpressing ACE2, were employed for these assays. In line with the biochemical results, MAVDA-B-219 showed the most potent inhibition of SARS-CoV-2 viral replication ($IC_{50}$ values of 14.9 and 16.3 μM in A549 and Huh7 cells, respectively) (Fig. 5D; Appendix Fig. S16). No apparent cytotoxicity was seen for any of the best-performing derivatives up to a concentration of 100 μM (Appendix Fig. S17).

# Discussion

Here, we report a cell-based multiplex drug screening platform capable of rapidly detecting chemical inhibitors across multiple targets simultaneously. Given the persistent threat posed by emerging pathogens and the critical need for pandemic preparedness, we selected viral proteases as a well-established and relevant family of therapeutic targets. Shifting away from the conventional single-target drug screening paradigm, our scalable multiplex approach enables expedited assessment of chemical interactions, significantly reducing the time, effort, and resources required. Moreover, it effectively identifies broad-acting inhibitors early in the screening process, which are particularly valuable in the context of evolving viruses and future risks. The advantages of this approach are evident in our initial screen, which involved testing a modest collection of 2480 compounds against a pool of 40 viral proteases—derived primarily from pathogens that lack approved antivirals and several of which have been identified as posing high risk to public health— as well as several non-protease controls. This screen yielded 106,640 total test combinations, uncovering 48 verified compound-target interactions across models belonging to coronavirus, picornavirus, and retrovirus families. Additionally, we identified a series of broad-acting lead candidates, which were validated as true hits across orthogonal assays, demonstrating the potential of multiplex screens to rapidly identify lead compounds against a broad range of viral targets. Furthermore, when looking at the strength of our identified hits, they show similar potency to the initial hits obtained from classic chemical screens (Lim et al, 2021; Ma et al, 2021; Wang et al, 2022; Zang et al, 2023; W. Zhu et al, 2020). Overall, these results suggest that our method, which utilizes extensive multiplexing and yeast as biosensors to monitor protease activity, yielded compounds with properties comparable to those obtained through more conventional approaches.

There are several innovations that contribute to the versatility and robustness of our screening platform, including the use of yeast as a tunable cellular biosensor for protease activity, DNA-barcoding to simultaneously track multiple models at once, and redundant barcoding with multiple in-well controls to increase assay specificity and sensitivity. The choice of yeast as a model system is particularly advantageous due to its cost-effectiveness, scalability from its ability to grow in suspension, rapid growth rate, and ease of genetic engineering, allowing for the generation of hundreds of clonal barcoded lines in a short time frame. As demonstrated by our investigation on viral proteases along with human kinase

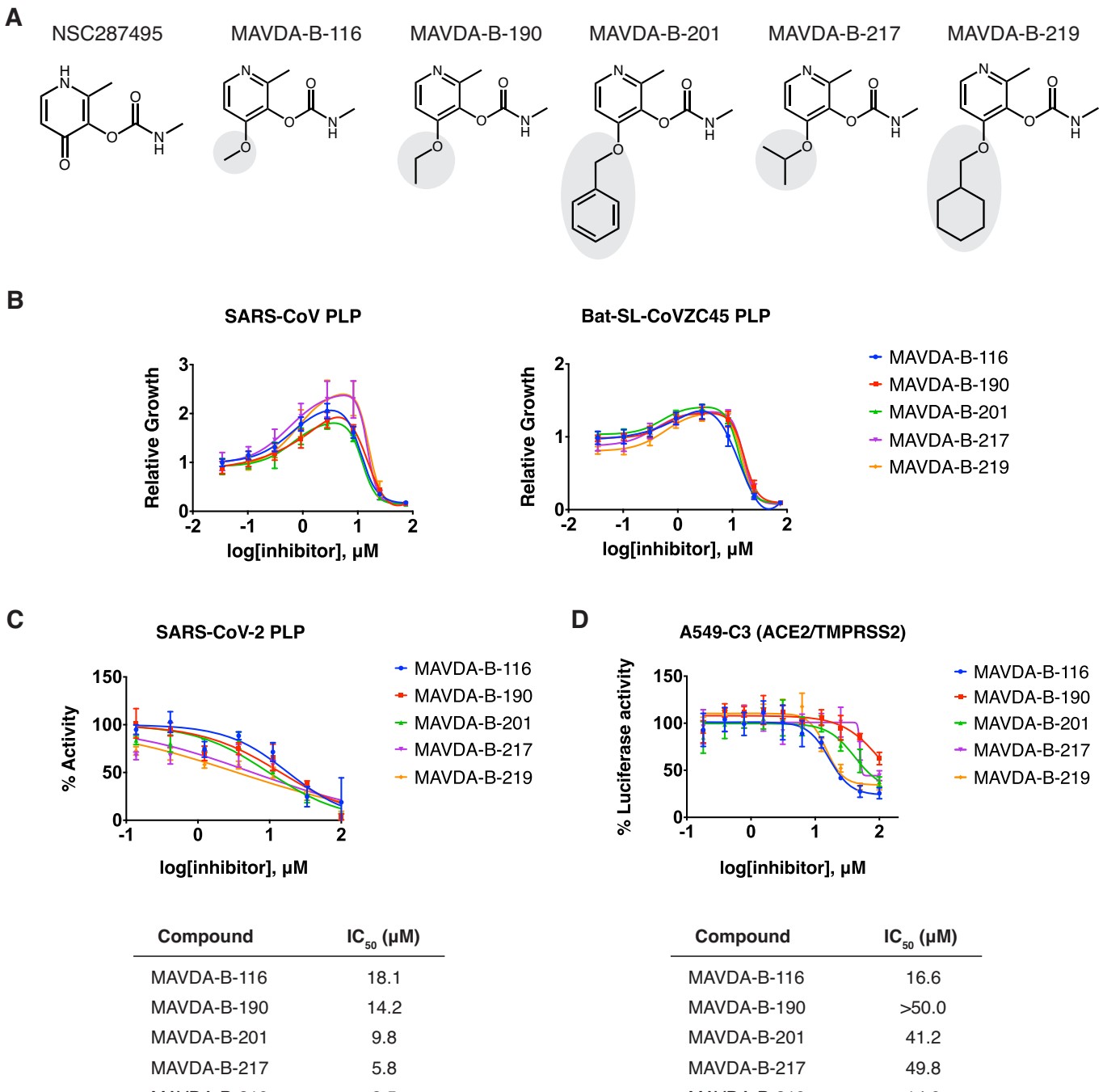

**Figure 5. Broad-acting coronavirus PLP inhibitors.**

(A) Structures of NSC287495 and analogs, MAVDA-B-116, MAVDA-B-190, MAVDA-B-201, MAVDA-B-217, and MAVDA-B-219. (B) Dose-dependent rescue of growth in *S. cerevisiae* expressing SARS-CoV PLP or Bat SL-CoVZC45 PLP by corresponding analogs and parent compound. Error bars denote the mean ± s.d. of five biological replicates. The x-axis is in the $\log_{10}$ scale. (C) Inhibition of purified SARS-CoV-2 PLP by the analogs. Error bars denote mean ± s.e.m. of three technical replicates. Nonlinear regression was used to determine the $IC_{50}$s. The x-axis is in the $\log_{10}$ scale. (D) Inhibition of SARS-CoV-2 viral replication in A549-C3 cells. Error bars denote mean ± s.d. of four technical replicates. Nonlinear regression was used to determine the $IC_{50}$s. The x-axis is in $\log_{10}$ scale. Source data are available online for this figure.

controls, an additional advantage of our approach includes the ability to assess the activity of proteins within a complex intracellular milieu, bypassing the need for costly protein purification as would be required by alternative screening methods, or

stringent biosafety procedures associated with live viruses. Moreover, the platform's flexibility allows for easy addition or removal of models as needed through a simple reassembly of the initial starting pool. Our hit-calling methodology, which relies on the redundant

barcodes and behavior of the various in-well controls built into our pool, consistently identified reproducible hits, with those with the highest magratios showing concordant increases in their rate of validation. Using a magratio cutoff of 0.3, where 1 represents the activity of a screening compound on par with the strong positive control, lopinavir, enabled us to achieve a precision of 0.80. Should future groups desire to capture more potential hits, a lower magratio cutoff could be applied, understanding that while this would increase sensitivity, it would also increase the number of false positives. The use of the more classic z-prime metric (Zhang et al, 1999) was also explored during our studies, but found to be unsuitable for accurate hit calling (see Appendix Supplementary Note S1 for additional discussion).

While the primary focus of this work is the establishment and rigorous validation of a multiplexed screening approach, several interesting screening hits were subjected to more in-depth follow-up studies. Among these, we further examined a series of coronavirus inhibitors. The three chromen-2-one analogs, CB6778425, CB6728297, and CB6762077, exhibited broad activity against the panel of coronavirus 3CL proteases, and subsequent analyses of these hits using an orthogonal biochemical assay confirmed the target-specific activity of each against the SARS-CoV-2 3CL protease. For the PLP inhibitor, by generating and testing a variety of analogs, a set of improved variants were identified, which were validated for activity in vitro using purified protein as well as authentic SARS-CoV-2. In examining the resulting data using the purified SARS-CoV-2 PLP enzyme, a distinct difference in the $IC_{50}$ values between the analogs was observed (ranging from 3.5 to 14.2 $\mu$M), whereas the range in $EC_{50}$ values against the yeast models was narrower (ranging from 0.7 to 0.9 $\mu$M) for all four highly active compounds. It is possible that compound toxicity is preventing us from observing the maximal amount of rescue within yeast and thus lowering the resulting $EC_{50}$ values we observe by causing the rescue to "prematurely peak" before declining due to toxicity. In addition, differences in binding of the analogs to the SARS-CoV-2 PLP as compared to the SARS-CoV PLP and Bat-CoVZC45 PLP may underlie the differences between the yeast and purified protein data. It is also noteworthy that during the course of our validation studies, the relatively weak response of NSC287495 against the SARS-CoV-2 PLP yeast model precluded us from using it for hit optimization. This lack of sensitivity in the SARS-CoV-2 PLP model to chemical inhibitors appears to be a property of the specific model. This conclusion is supported by experiments testing the published inhibitor, GRL0617, which shows similar in vitro activity against purified SARS-CoV PLP and SARS-CoV-2 PLP enzymes, yet against the SARS-CoV-2 yeast model shows a blunted response as compared to the SARS-CoV model (Appendix Fig. S18). Future studies will be required to better understand how to further tune each of the protease models to maximize their sensitivity.

Lastly, while our study focused on validating broad-acting viral inhibitors, as these have the potential to treat many current infectious agents along with aiding in pandemic preparedness, other domains may require selective modulators. This need highlights another valuable function of the multiplex screen. For example, precision medicine often aims to target a single protein within a broad family of structurally similar members (e.g., kinases) (Müller et al, 2015), which makes selective modulation challenging. In these scenarios, our platform can help by not only identifying

compounds that hit a target protein, but also identifying compounds that hit related, off-targets. While not the focus of this study, we uncovered a number of putative protease inhibitors that showed target selectivity in our screens (Fig. 3A; Appendix Fig. S7). Additionally, we identified putative kinase inhibitors with selective activity, which suggests that our platform can be applied to targets outside of viral proteases, although further testing with orthogonal assays will be required to verify the activity of these hits.

Here, using a multiplex cell-based drug screening platform, we demonstrate the possibility of identifying inhibitors to a range of viral proteases as well as human proteins at an accelerated pace relative to conventional approaches. Furthermore, while this approach makes use of next-generation sequencing, the added cost per compound is ~$1/well screened (see Appendix Supplementary Note S2 for details), with the bulk of those costs incurring from the short-read sequencing and not the sample preparation. As the costs of sequencing continue to decrease, we anticipate substantial reductions in the cost per sample analyzed. In future efforts, the platform can be enhanced by increasing the size of the screening pool to enable more targets to be examined at once. Indeed, analysis of the multiplexing capacity of our screen, through simulation of reduced sequencing depth, revealed that the assay may be scaled by approximately five-fold without further modification (Appendix Fig. S19). In addition, by adding barcoded yeast strains deficient in key biological processes and observing their depletion within the pool, it may be possible to profile compounds for toxicity due to interactions with core biological pathways during the initial screening process. While we employed the drug-sensitized pdr1$\Delta$ prd3$\Delta$ snq2$\Delta$ strain (Piotrowski et al, 2017), which has been well documented to drastically increase small molecule permeability and retention, additional genes involved in regulating small molecule efflux such as YRR1 or PDR8 can also be removed to help further improve small molecule accumulation in yeast. The potential of alternative cellular chassis systems, such as suspension cultures of mammalian cells like CHO or HeLa-S3, may also be considered. Furthermore, continued efforts to refine the data analysis pipeline will invariably improve assay sensitivity while maintaining its high specificity to help minimize the amount of subsequent validation required. Finally, while we have selected cytotoxicity as the basis of our screen, future iterations of this platform can be designed to incorporate more sophisticated sensors of enzyme activity such as synthetic circuits that report protein activity. Adapting distinct sensors may also enable protein targets whose expression alone does not cause cytotoxicity to be screened within this platform.

## Methods

**Reagents and tools table**

| Reagent/resource | Reference or source | Identifier or catalog number |
| --- | --- | --- |
| **Experimental models** | | |
| BY4741 (*S. cerevisiae*) | Gift of Dr. F. Bradley Johnson (UPenn) | |
| Huh7-ACE2 | Liu et al, 2022 | |
| A549-ACE2plusC3 | Chang et al, 2022 | |

| Reagent/resource | Reference or source | Identifier or catalog number |
|---|---|---|
| **Recombinant DNA** | | |
| Yeast expression vectors | This study | Table EV1 |
| **Oligonucleotides and other sequence-based reagents** | | |
| PCR primers | This study | Table EV2 |
| **Chemicals, enzymes, and other reagents** | | |
| NCI Diversity Set VI | NIH DSCB | |
| DIVERSet-EXP library | Chembridge | NS1519 |
| Gateway BP Clonase II Enzyme mix | Invitrogen | Cat # 11789100 |
| Gateway LR Clonase II Enzyme mix | Invitrogen | Cat # 11791100 |
| SacI-HF | New England Biolabs | R3156S |
| NgoMIV | New England Biolabs | R0564S |
| SARS-CoV-2 3CL | Iketani et al, 2021 | |
| SARS-CoV-2 PLP | Acro Biosystems | PAE-C5184 |
| MCA- AVLQSGFR-Lys(DNP)-Lys-NH2 | GL Biochem | Cat # CV-070 |
| Z-Arg-Leu-Arg-Gly-Gly-AMC | Bachem | I1690 |
| Dabcyl-FRLKGGAPIKGV(EEdans)-NH2 | Biosynth Ltd (US) | |
| **Software** | | |
| BaseSpace Sequence Hub | Illumina | |
| Magellan | Tecan | |
| Gen5 | Agilent | |
| SoftMax Pro v7.0.2 | Molecular Devices | |
| flexControl / flexAnalysis v3.4 | Bruker | |
| Prism v10.0.3 | GraphPad | |
| Excel v16.82 | Microsoft | |
| **Other** | | |
| Illumina NextSeq 500/550 | Illumina | |
| Infinite F50 | Tecan | |
| BioTek Synergy HTX | Agilent | |
| BioTek Synergy Neo2 | Agilent | |

## Yeast and media

The *S. cerevisiae* yeast strain BY4741 (MATa his3Δ1 leu2Δ0 ura3Δ0 met15Δ0) with drug-sensitizing mutations (pdr1Δ prd3Δ snq2Δ) (Piotrowski et al, 2017) was used for the construction of barcoded protease-expressing yeast models. The parental strain was first transformed with protease expression vectors and subsequently with barcode plasmids using a modified lithium acetate-heat shock method (Gietz and Schiestl, 2007). Briefly, yeast strains were grown to saturation overnight in YPD (yeast extract-peptone-dextrose) at 30 °C and then diluted 1:100 in YPD (~2.5 mL per transformation) and grown for 4 h prior to

transformation to achieve log-phase growth. For each transformation, plate mix comprising 240 μL 50% Peg3350, 30 μL 10X TE (100 mM Tris-HCL, 10 mM EDTA), 5 μL 10 mg/mL ssDNA, 30 μL 1 M LiOAc and 34 μL DMSO was combined with 200–300 ng expression vector or barcode plasmid and 50 μL of yeast resuspended in 1X LiAc/TE (100 mM LiAC, 10 mM Tris-HCl, 1 mM EDTA), and incubated at 42 °C for 20 min without shaking. Cells were then washed with 1 mL sterile PBS, and resuspended in 100 μL of PBS to be plated onto appropriate selection media and grown for 48 h at 30 °C.

The yeast media used in this study were prepared as follows: YPD consisted of 10 g/L yeast extract, 20 g/L peptone, and 20 g/L D-(+)-glucose. Synthetic complete (SC) media used for the maintenance of strains that harbored the protease expression vectors and/or barcode plasmids comprised 1.5 g/L drop-out mix lacking histidine, leucine, and uracil without yeast nitrogen base (US Biological), 1.7 g/L yeast nitrogen base without amino acids, carbohydrate, and without ammonium sulfate (US Biological), 5 g/L ammonium sulfate, 20 g/L D-(+)-glucose (GLU) or galactose (GAL), and was supplemented to 90 mg/L histidine and 180 mg/L leucine as required. Agar plates were prepared using the same recipe with the addition of 2% (w/v) agar and 600 μL of 5 M NaOH per liter to help the plate solidify well.

## Plasmids

Viral protease genes were synthesized with the addition of a start codon, ATG, at the 5′ end and a stop codon, TAA, at the 3′ end (Twist Bioscience) and cloned into pDONR221 using Gateway BP Clonase II Enzyme mix (Invitrogen) to produce entry vectors. The entry vector encoding the human kinase, YES1, was obtained from the hOrfeome V8.1 Library collection, and those for c-Src and FES were cloned from Addgene plasmids #44652 (Ogura et al, 2012) and #23876 (Johannessen et al, 2010), respectively. Entry vectors were then cloned into various destination vectors using Gateway LR Clonase II Enzyme mix (Invitrogen). Destination vectors that were used in this study include pAG426GAL-ccdB (Addgene plasmid #14155), pAG416-GAL-ccdB (Addgene plasmid #14147), pAG416-GAL10p-ccdb-6Stop, pAG416-GALL-ccdB-6stop, pAG416-GAL-1X-ccdB-6stop, pAG416-GAL-2X-uA-ccdB-6stop, and pAG416-GAL-3X-uC-ccdB-6stop (1X to 3X refers to the number of Gal4 binding sites within the promoter, and the letter following the "u" refers to the base modification at a critical position within the Kozak sequence upstream of the initiating ATG). Plasmids used in this study have been posted to Addgene (See Table EV1 for details).

Barcode plasmids were constructed using pAG415GAL-ccdB (Addgene plasmid #14145) as a backbone vector. Briefly, the vector was digested using SacI and NgoMIV, and the resulting ~2800 bp fragment was replaced with the kanamycin resistance gene followed by a 10 bp barcode via ligation.

Plasmids encoding the protein sequences were isolated using a standard miniprep protocol (Omega Biotek, Biobasic), and barcode plasmids were isolated using the Zyppy-96 Plasmid Kit (Zymo Research). For plasmid verification, Sanger sequencing (Genewiz) was used to confirm the genes, and next-generation sequencing to identify the DNA barcodes (Illumina). A set of barcode sequences that were determined to be at least 3 hamming distances apart were selected and used to construct the yeast models.

## Yeast spot assays

Yeast strains were grown to saturation over two days in non-inducing media (SC -ura GLU) and serially diluted 1:10 in sterile PBS (Gibco). About 5 μL of each dilution were spotted onto agar plates containing non-inducing (SC -ura GLU) or inducing (SC -ura GAL) media. Plates were incubated for 48 h at 30 °C before being imaged.

## Multiplex screening

To prepare the screening pool, 220 barcoded yeast strains, each harboring a viral protease or control expression vector, were individually grown to saturation in non-inducing media (SC -ura -leu GLU) and mixed in equal volumes. The resulting pool was mixed thoroughly and aliquoted into 300 μL volumes, and each aliquot was mixed with 200 μL of 50% glycerol for storage at −80 °C. For each round of screening, a frozen vial of the yeast pool glycerol stock was thawed and inoculated into 6 mL of non-inducing media (SC -ura -leu GLU) overnight to prepare the starter culture for screening.

Drug screens were conducted in 96-well deep-well plates (VWR) using 1 mL of inducing media (SC -ura -leu GAL) per well. The yeast starter culture was inoculated at 1:1000, and 2.5 μL of compound or DMSO were added to each well. Compound stocks were prepared at 10 mM concentration resulting in a final screening concentration of 25 μM. The plates were then grown for 40 h at 30 °C with shaking at 1000 rpm.

After growth, the optical density (OD595) of the culture was measured using a 96-well plate reader (Tecan) by taking 100 μL from each well. The remaining culture was then processed for DNA extraction using a modified LiOAc-SDS lysis method as done previously (Resnick et al, 2022).

## Compound libraries

The yeast pool was screened against 1520 compounds from the NCI Diversity Set VI (https://dtp.cancer.gov/organization/dscb/obtaining/available_plates.htm) and 960 compounds from the DIVERSet-EXP library (Chembridge).

## Sequencing library preparation

To prepare sequencing libraries, the barcode pools extracted from each well were amplified and uniquely indexed via a single round of PCR using primers that were designed to contain the indexed sequencing adapter, an internal primer barcode (forward primer only), and a common priming site for the yeast plasmid barcode (see Table EV2 for primers used). Each reaction consisted of 2 μL 10X Taq buffer, 0.4 μL 10 mM dNTPs, 1 μL 10 μM forward primer, 1 μL 10 μM reverse primer, 0.5 μL template DNA, 0.1 μL Taq polymerase (Enzymatics), and 15 μL $H_2O$. The cycle conditions were as follows:

1. 94 °C, 3 min
2. 94 °C, 30 s
3. 57 °C, 20 s
4. 72 °C, 30 s
5. Return to step 2, 23 X
6. 72 °C, 3 min
7. Hold at 4 °C

Each PCR was conducted in technical duplicate and combined, and all reaction products from a screened plate were pooled together prior to being run on a gel. The band corresponding to the correct amplicon size (336 bp) was gel purified, and the samples to be sequenced were quantified using the Collibri Library Quantification Kit (Thermo Fisher Scientific). Pooled libraries were sequenced using the NextSeq 500/550 platform (Illumina) with 75 cycles.

## Analysis of multiplex screening data

Each screened plate consisted of a set of DMSO wells (negative controls; 14 wells) and compound wells (2 lopinavir positive controls and 80 random library compounds). Each well included the same pool of 220 barcoded strains, including the protease models, eYFP negative control, and kinase models. To identify compounds that rescue yeast growth, barcode read counts were used for the analysis and these were processed on a per-plate basis to account for batch effects.

First, wells were high-pass filtered by read counts to ensure sufficient coverage, only retaining wells that had at least 30,000 total reads (Appendix Fig. S5B,C). Next, a high-pass filter was applied to all wells by optical density (OD) at 595 nm; all wells with an OD595 of at least 0.40, or that were at most half a standard deviation below the mean OD of the plate were retained. In this manner, the optical density threshold was dynamic to allow plates with overall lower growth to still be processed.

After filtering wells based on reads and optical density, well counts were sampled with replacement to create 1000 pseudowell counts, where each pseudowell contained one barcoded strain per model. To determine growth relative to the negative control model (EYFP), the pseudowell reads were normalized by dividing each barcode read count by the EYFP read count of the respective pseudowell. When the mean pseudowell read count for a model was greater in DMSO than for a given compound, a $p$ value of 1 was imputed for the model-compound interaction. Otherwise, the interaction between a given model and compound was assessed by bootstrapping 1000 independent $t$-tests to compare the compound pseudowells and the DMSO pseudowells. Five random samples (the number of barcodes per model) from the compound pseudowell read counts and a proportional number of counts from the DMSO pseudowells were used in each $t$-test. Subsequently, the geometric mean of the resulting $p$ values was calculated to produce a single average $p$ value for each model-compound interaction.

To remove spurious hits, two normalization factors were included. The first factor aimed to make hits on kinases and proteases mutually exclusive within each well, as it was assumed that compounds may inhibit proteases or kinases but not both. To calculate this factor for a given well, the minimum $p$ value of each protease model and of each kinase model in the well was identified, and the larger of these two minimums was selected as the first factor. The second factor aimed to limit the number of hits per model across all wells within the plate, as hits were assumed to be rare, and the first quartile of $p$ values across all wells for a given model was used for this purpose. All $p$ values from the analysis were divided by the corresponding two normalization factors, resulting in the normalized $p$ values.

Finally, the magratio score for a model-compound interaction was calculated as the normalized $p$ value relative to the best-normalized $p$ value among positive controls. Specifically, the magratio for an interaction was calculated as the logarithm of the respective interaction's normalized $p$ value divided by the logarithm of the lowest normalized $p$ value across the positive controls. When the logarithm of the smallest normalized $p$ value among positive controls was greater than $-10$, the positive controls were considered to have failed, and the given plate was not further evaluated.

For DMSO and Lopinavir control wells, we further examined the coefficient of variation and variance of our well-normalized control gene reads across all barcodes across all plates, and observed unimodal distributions with minor to moderate skew, respectively, suggesting that our data is homoscedastic (Appendix Fig. S20).

## Yeast dose-response liquid growth assay

Barcoded yeast strains for each test compound were grown to saturation over two days in 1 mL of non-inducing media (SC -ura -leu GLU) at 30 °C with shaking at 1000 rpm. Subsequently, strains were inoculated into 1 mL of inducing media (SC -ura -leu GAL) with the test compound at eight concentrations ranging from 34.3 nM to 75.0 μM. To accommodate slight variations in the growth rates across the yeast models, and to prevent saturation of the liquid culture, which limits the dynamic range of the assay, strains were either inoculated at 1:500 and grown for 42–48 h or, inoculated at 1:100, grown for 24 h, and passaged at 1:100 or 1:250 into the same conditions and grown for an additional 24 h. All cultures were grown at 30 °C with shaking at 1000 rpm prior to measuring the OD595 of each well by taking 100 μL and using a plate reader (Tecan). The OD values were normalized against the average of values measured at the lowest drug concentration to determine relative growth, and EC50 values were determined by nonlinear regression (GraphPad Prism). A hit was identified as a true positive if the relative growth in the target model between the lowest and highest tested concentrations (without toxicity) was greater than 1.2X, with statistical significance calculated by an unpaired, one-tailed $t$-test ($p < 0.05$) using the corresponding values (Table EV3).

## Purification of SARS-CoV-2 3CL protease

SARS-CoV-2 3CL protease was purified as previously described (Iketani et al, 2021). Briefly, BL21 (DE3) cells transformed with pGEX-5X-3-SARS-CoV-2-3CL (Addgene #168457) were grown at 37 °C, 220 RPM until OD600 equals 0.6–0.7, and then induced by addition of 0.5 mM IPTG and further incubated at 16 °C, 180 RPM for 10 h. Cells were pelleted, resuspended in 20 mM Tris-HCl, pH 8.0, 300 mM NaCl, homogenized by sonication, clarified by centrifuging at $25,000 \times g$ for 1 h, and then the supernatant was incubated with Glutathione Sepharose resin (Sigma) for 2 h at 4 °C. The resin was then extensively washed and incubated with Factor Xa for 36 h at 4 °C, then purified by size exclusion chromatography using a Superdex 10/300 GL column (Cytiva) in 50 mM Tris-HCl, pH 7.5, 1 mM EDTA. The quality of the purified protease was validated by SDS-PAGE and measurement of biochemical activity.

## Measurement of biochemical inhibition of peptide cleavage by SARS-CoV-2 3CL protease

The in vitro biochemical activity of the SARS-CoV-2 3CL protease was measured as previously described (Iketani et al, 2021). The fluorogenic peptide substrate MCA-AVLQSGFR-Lys(DNP)-Lys-NH2, corresponding to the nsp4/nsp5 cleavage site in the virus, was synthesized (GL Biochem) and resuspended in DMSO. For the measurement of IC50 values, in a 96-well plate, the protease in the assay buffer (50 mM Tris-HCl, pH 7.5, 1 mM EDTA) was first added to each well at a final concentration of 0.2 μM. Serial dilutions of the test compound were prepared in the assay buffer. The substrate was then added at 20 μM, and then fluorescence (Excitation 320 nm, Emission 405 nm) at 30 °C was continuously measured on a plate reader for 20 min. Inhibition was calculated by comparison to control wells with no added inhibitor (negative control) and with no added protease (positive control). IC50 values were determined by fitting data with Log(inhibitor) vs. normalized response curve (the standard inhibition curve, GraphPad Prism).

## Measurement of biochemical inhibition of peptide cleavage by SARS-CoV-2 PLP

Recombinant SARS-CoV-2 PLP was purchased from Acro Biosystems (PAE-C5184) and diluted to 20 nM in assay buffer consisting of 50 mM HEPES pH 6.5, 150 mM NaCl, 0.01% Tween-20, and 0.1 mM dithiothreitol. Compounds stored as 10 mM stocks in DMSO, were threefold serially diluted in DMSO (six times) to 13.72 μM. From here, compounds were further diluted 1 in 25 in assay buffer and then incubated with an equal volume of 20 nM of SARS-CoV-2 PLP for 30 min on ice. The reaction was started by the addition of 50 μM of Z-Arg-Leu-Arg-Gly-Gly-AMC (Bachem, I1690). All assays were performed in triplicate wells on a 384-well plate, and the final concentrations of enzyme and substrate in each well were 5 nM and 25 μM, respectively. The final inhibitor concentration ranged from 100 μM to 137 nM. Wells were incubated for 1 h at 37 °C in a Synergy HTX (Biotek) plate reader and readings were recorded at an excitation wavelength of 360 nm and emission wavelength of 460 nm. The highest velocity of each reaction was monitored over 12 sequential readings and recorded as relative fluorescent units per sec. The velocity of each reaction was normalized to the DMSO control. Dose-response curves were generated using GraphPad Prism.

## SARS-CoV-2 PLP time-dependent dose-response curve (DRC) assay

Experiments were conducted using an assay buffer containing 50 mM Tris (pH 7.3), 1 mM EDTA, 0.01% Tween-20, and 1 mM TCEP. Compounds were tested at final concentrations ranging from 0.012 to 100 μM. The SARS-CoV-2 PLP enzyme was added at a final concentration of 100 nM and preincubated for varying durations of 10–120 min. After pre-incubation, the substrate, Dabcyl-FRLKGGAPIKGV(EEdans)-NH2 was added at a final concentration of 40 μM. Reactions were carried out in a total volume of 30 μL in 384-well, non-binding, black plates (Greiner Bio-one). After 90 min of incubation at room temperature, fluorescence intensity was measured at an excitation wavelength

of 360 nm and an emission wavelength of 450 nm using a Neo®2 microplate reader (Agilent).

### SARS-CoV-2 PLP X100 assay

The same reagents were used as in the SARS-CoV-2 PLP time-dependent DRC assay. A 10 µL mixture of 10 µM enzyme and 10 µM compound was preincubated for 120 min. Subsequently, 1 mL of 40 µM substrate was added, diluting the assay 100-fold. After 90 min of incubation at room temperature, a 30 µL sample was transferred to a 384-well plate and read as described in the previous experiment.

### Mass spectrometry analysis of protein-inhibitor complex

To demonstrate the binding of covalent inhibitors, 50 µM of the SARS-CoV-2 3CL protease was incubated with 500 µM of inhibitor in a buffer comprising 50 mM Tris-HCl (pH 7.5), 1 mM EDTA for 1 h at 4 °C to acquire the protein-inhibitor complex before analysis by Mass Spectrometry. For MALDI-TOF MS analysis, 1 µL of the protein-inhibitor complex (3CL protease only and 3CL protease-CB6778425) was mixed with 9 µL of 10 mg/mL sinapinic acid in the matrix solution (70:30 water/acetonitrile, with 0.1% TFA). About 1.0 µL of the final mix was deposited onto the target carrier and allowed to air dry. MALDI spectra of the protein-inhibitor complex were compared with ligand-free 3CL protease to determine the mass shift. MALDI spectra were collected with the software flexControl, while MALDI data were analyzed by flexAnalysis.

### Live virus testing

To characterize the IC$_{50}$ value of each drug against the live SARS-CoV-2 virus, Huh7-ACE2 (Iketani et al, 2022; Liu et al, 2022) and A549-ACE2plusC3 (Chang et al, 2022) cells were seeded at a density of $10^4$ cells per well of a 96-well plate. The following day, the cells were infected with the SARS-CoV-2 nLuc reporter virus (USA-WA1/2020 strain with ORF7a replaced by NanoLuc) (Iketani et al, 2022; Ye et al, 2020) at an MOI of 0.05 and treated with the indicated drugs in a twofold dilution series. At 24 h post infection, the cells were lysed, and luminescence activity was assayed using the Promega Nano-Glo Luciferase Assay System (Cat# N1120). All experiments were done in quadruplicates. IC$_{50}$ values were derived by fitting a nonlinear regression curve to the data in GraphPad Prism.

To assess cytotoxicity of each compound, Huh7-ACE2 and A549-ACE2plusC3 cells were seeded at a density of $10^4$ cells per well of a 96-well plate. The following day, indicated compounds were added to the cultures in a twofold dilution series. Forty-eight hours after the addition of the compounds, cytotoxicity was measured using Promega CellTiter-Glo® 2.0 Cell Viability Assay (Cat# G9243). All experiments were done in quadruplicates. CC$_{50}$ values were derived by fitting a nonlinear regression curve to the data in GraphPad Prism. All cell lines were purchased from authenticated vendors (authentication included morphology check under microscopes and growth curve analysis) and tested mycoplasma negative.

## Data availability

The datasets and computer code produced in this study are available in the following databases:Raw sequencing data: NCBI SRA BioProject PRJNA901605. Raw data tables and computer scripts: GitHub (https://github.com/alejandrochavezlab/mavda). Computational code: Zenodo (https://doi.org/10.5281/zenodo.13858988.

The source data of this paper are collected in the following database record: biostudies:S-SCDT-10_1038-S44320-024-00082-1.

## Peer review information

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

## Acknowledgements

Funding for this study was provided by NIAID (1U19AI1711401) to JMB, YS, JSF, DDH, and AC. AC is additionally supported by a Career Awards for Medical Scientists from the Burroughs Wellcome Fund (1017195.01) and a grant from the Jack Ma Foundation. The National Cancer Institute (NCI)/Division of Cancer Treatment and Diagnosis (DCTD)/Developmental Therapeutics Program (DTP) provided the Diversity Set VI compounds (http:// dtp.cancer.gov). BAA is supported by the National Science Foundation Graduate Research Fellowship Program under Grant No. DGE-2038238. Any opinions, findings, conclusions, or recommendations expressed in this material are those of the authors and do not necessarily reflect the views of the National Science Foundation.

## Author contributions

**Seo Jung Hong**: Data curation; Formal analysis; Validation; Investigation; Visualization; Methodology; Writing—original draft; Writing—review and editing. **Samuel J Resnick**: Data curation; Formal analysis; Validation; Investigation; Methodology; Writing—review and editing. **Sho Iketani**: Data

curation; Formal analysis; Validation; Investigation; Methodology; Writing—review and editing. **Ji Won Cha**: Data curation; Formal analysis; Validation; Investigation; Writing—review and editing. **Benjamin Alexander Albert**: Data curation; Software; Formal analysis; Investigation; Visualization; Methodology; Writing—review and editing. **Christopher T Fazekas**: Data curation; Formal analysis; Validation; Investigation; Visualization; Writing—review and editing. **Ching-Wen Chang**: Formal analysis; Validation; Visualization; Writing—review and editing. **Hengrui Liu**: Formal analysis; Validation; Visualization; Writing—review and editing. **Shlomi Dagan**: Formal analysis; Validation; Visualization; Writing—review and editing. **Michael R Abagyan**: Formal analysis; Validation; Visualization; Writing—review and editing. **Pavla Fajtová**: Formal analysis; Validation; Visualization; Writing—review and editing. **Bruce Culbertson**: Formal analysis; Validation; Writing—review and editing. **Brooklyn Brace**: Validation; Methodology. **Eswar R Reddem**: Formal analysis; Validation. **Farhad Forouhar**: Investigation. **J Fraser Glickman**: Formal analysis; Visualization; Writing—review and editing. **James M Balkovec**: Investigation; Methodology. **Brent R Stockwell**: Supervision; Validation. **Lawrence Shapiro**: Supervision; Validation. **Anthony J O'Donoghue**: Formal analysis; Supervision; Validation; Writing—review and editing. **Yosef Sabo**: Formal analysis; Supervision; Validation; Writing—review and editing. **Joel S Freundlich**: Supervision; Investigation; Visualization; Methodology; Writing—review and editing. **David D Ho**: Supervision; Investigation; Writing—review and editing. **Alejandro Chavez**: Conceptualization; Supervision; Funding acquisition; Investigation; Methodology; Writing—original draft; Writing—review and editing.

Source data underlying figure panels in this paper may have individual authorship assigned. Where available, figure panel/source data authorship is listed in the following database record: biostudies:S-SCDT-10_1038-S44320-024-00082-1.

## Disclosure and competing interests statement

AC, SJH, and SJR have submitted a patent related to the design and implementation of multiplex small molecule screens. BRS is an inventor on patents and patent applications involving small molecules, holds equity in and serves as a consultant to Exarta Therapeutics and ProJenX Inc., holds equity in Sonata Therapeutics, and serves as a consultant to Weatherwax Biotechnologies Corporation and Akin Gump Strauss Hauer & Feld LLP. DDH is a co-founder of TaiMed Biologics and RenBio, consultant to WuXi Biologics, Brii Biosciences, Apexigen, and Veru Inc., and board director for Vicarious Surgical.

