## [Peer Review File · Molecular Systems Biology]

A multiplex method for rapidly identifying viral protease inhibitors

Seo Jung Hong, Samuel Resnick, Sho Iketani, Ji Won Cha, Benjamin Albert, Christopher Fazekas, Ching-Wen Chang, Hengrui Liu, Shlomi Dagan, Michael Abagyan, Pavla Fajtová, Bruce Culbertson, Brooklyn Brace, Eswar Reddem, Farhad Forouhar, J. Fraser Glickman, James Balkovec, Brent Stockwell, Lawrence Shapiro, Anthony O'Donoghue, Yosef Sabo, Joel Freundlich, David Ho, and Alejandro Chavez

Corresponding author(s): Alejandro Chavez (chavez2@health.ucsd.edu)

Review Timeline:

Submission Date:	9th Jun 24
Editorial Decision:	9th Jul 24
Revision Received:	8th Oct 24
Editorial Decision:	11th Nov 24
Revision Received:	30th Nov 24
Accepted:	2nd Dec 24

Editor: Jingyi Hou

Transaction Report:

9th Jul 2024

Manuscript Number: MSB-2024-12417

Title: A multiplex method for rapidly identifying viral protease inhibitors

Author: Alejandro Chavez

Seo Jung Hong

Samuel Resnick

Sho Iketani

Ji Won Cha

Benjamin Albert

Christopher Fazekas

Ching-Wen Chang

Hengrui Liu

Michael Abagyan

Pavla Fajtová

Bruce Culbertson

Brooklyn Brace

Eswar Reddem

Farhad Forouhar

James Balkovec

Brent Stockwell

Lawrence Shapiro

Anthony O'Donoghue

Yosef Sabo

Joel Freundlich

David Ho

Dear Alex,

Thank you for submitting your work to Molecular Systems Biology. We have now heard back from the two reviewers who agreed to evaluate your manuscript. As you will see from the reports below, the reviewers are overall supportive. They raise, however, a series of concerns, which we would ask you to address in a major revision.

The reviewers' recommendations are relatively clear, so there is no need for me to reiterate the points listed below. All the issues raised by the reviewers need to be satisfactorily addressed. As you may already know, our editorial policy allows in principle a single round of major revision, and it is therefore essential to provide responses to the reviewers' comments that are as complete as possible.

On a more editorial level, we would ask you to address the following issues:

- Please provide a .docx formatted version of the manuscript text (including legends for main figures, EV figures and tables). Please make sure that the changes are highlighted to be clearly visible.
- Please provide individual production quality figure files as .eps, .tif, .jpg (one file per figure).
- Please provide a .docx formatted letter INCLUDING the reviewers' reports and your detailed point-by-point responses to their comments. As part of the EMBO Press transparent editorial process, the point-by-point response is part of the Review Process File (RPF), which will be published alongside your paper.
- Please note that all corresponding authors are required to supply an ORCID ID for their name upon submission of a revised manuscript.
- We replaced Supplementary Information with Expanded View (EV) Figures and Tables that are collapsible/expandable online (see examples in <http://msb.embopress.org/content/11/6/812>). A maximum of 5 EV Figures can be typeset. EV Figures should be cited as 'Figure EV1, Figure EV2' etc... in the text and their respective legends should be included in the main text after the legends of regular figures.

Additional Tables/Datasets should be labeled and referred to as Table EV1, Dataset EV1, etc. Legends have to be provided in a separate tab in case of .xls files. Alternatively, the legend can be supplied as a separate text file (README) and zipped together with the Table/Dataset file.

For the figures and tables that you do NOT wish to display as Expanded View figures, they should be bundled together with their legends in a single PDF file called *Appendix*, which should start with a short Table of Content. Each legend should be below the corresponding Figure/Table in the Appendix. Appendix figures and tables should be referred to in the main text as: "Appendix Figure S1, Appendix Figure S2, Appendix Table S1" etc. See detailed instructions regarding expanded view here: <https://www.embopress.org/page/journal/17444292/authorguide#expandedview>.

-Before submitting your revision, primary datasets (and computer code, where appropriate) produced in this study need to be deposited in an appropriate public database (see [http://msb.embopress.org/authorguide - dataavailability](http://msb.embopress.org/authorguide-dataavailability) <https://www.embopress.org/page/journal/17444292/authorguide#dataavailability>).

The accession numbers and database should be listed in a formal "Data Availability" section (placed after Materials & Method) that follows the model below (see also <https://www.embopress.org/page/journal/17444292/authorguide#dataavailability>). Please note that the Data Availability Section is restricted to new primary data that are part of this study.

Data availability

-At EMBO Press we ask authors to provide source data for the main figures. Our source data coordinator will contact you to discuss which figure panels we would need source data for and will also provide you with helpful tips on how to upload and organize the files.

- Our journal encourages inclusion of *data citations in the reference list* to directly cite datasets that were re-used and obtained from public databases. Data citations in the article text are distinct from normal bibliographical citations and should directly link to the database records from which the data can be accessed. In the main text, data citations are formatted as follows: "Data ref: Smith et al, 2001". In the Reference list, data citations must be labeled with "[DATASET]". A data reference must provide the database name, accession number/identifiers and a resolvable link to the landing page from which the data can be accessed at the end of the reference. Further instructions are available at .

- We updated our journal's competing interests policy in January 2022 and request authors to consider both actual and perceived competing interests. Please review the policy <https://www.embopress.org/competing-interests> and update your competing interests if necessary.

Please use the heading "Disclosure statement and competing interests".

- All Materials and Methods need to be described in the main text using our 'Structured Methods' format, which is required for all research articles. According to this format, the Methods section includes a Reagents and Tools Table (listing key reagents, experimental models, software and relevant equipment and including their sources and relevant identifiers) followed by a Methods and Protocols section describing the methods using a step-by-step protocol format. The aim is to facilitate adoption of the methodologies across labs. More information on how to adhere to this format as well as a downloadable template (.docx) for the Reagents and Tools Table can be found in our author guidelines: <https://www.embopress.org/page/journal/17444292/authorguide#structuredmethods>.

An example of a Method paper with Structured Methods can be found here: <https://www.embopress.org/doi/10.15252/msb.20178071>.

-Regarding data quantification:

Please ensure to specify the name of the statistical test used to generate error bars and P values, the number (n) of independent experiments (please specify technical or biological replicates) underlying each data point and the test used to calculate p-values in each figure legend. Discussion of statistical methodology can be reported in the materials and methods section, but figure legends should contain a basic description of n, P and the test applied.

Graphs must include a description of the bars and the error bars (s.d., s.e.m.).

- Please provide a "standfirst text" summarizing the study in one or two sentences (approximately 250 characters, including space), three to four "bullet points" highlighting the main findings and a "synopsis image" (550px width and 400-600 px height, PNG format) to highlight the paper on our homepage.

Here are a couple of examples:

<https://www.embopress.org/doi/10.15252/msb.20199356>

<https://www.embopress.org/doi/10.15252/msb.20209475>

<https://www.embopress.org/doi/10.15252/msb.209495>

When you resubmit your manuscript, please download our CHECKLIST (<https://www.embopress.org/pb-assets/embosite/EMBO%20Press%20Author%20Checklist-1642513524327.xlsx>) and include the completed form in your submission.

Please note that the Author Checklist will be published alongside the paper as part of the transparent process (<https://www.embopress.org/page/journal/17444292/authorguide#transparentprocess>).

If you feel you can satisfactorily deal with these points and those listed by the referees, you may wish to submit a revised version of your manuscript. Please attach a covering letter giving details of the way in which you have handled each of the points raised by the referees. A revised manuscript will be once again subject to review and you probably understand that we can give you no guarantee at this stage that the eventual outcome will be favorable.

I look forward to receiving your revised manuscript soon.

Kind regards ,
Jingyi

Jingyi Hou, PhD
Scientific Editor
Molecular Systems Biology

We realize that it is difficult to revise to a specific deadline. In the interest of protecting the conceptual advance provided by the work, we recommend a revision within 3 months (7th Oct 2024). Please discuss the revision progress ahead of this time with the editor if you require more time to complete the revisions. Use the link below to submit your revision:

IMPORTANT: When you send your revision, we will require the following items:

1. the manuscript text in LaTeX, RTF or MS Word format
2. a letter with a detailed description of the changes made in response to the referees. Please specify clearly the exact places in the text (pages and paragraphs) where each change has been made in response to each specific comment given
3. three to four 'bullet points' highlighting the main findings of your study
4. a short 'blurb' text summarizing in two sentences the study (max. 250 characters)
5. a 'thumbnail image' (550px width and max 400px height, Illustrator, PowerPoint or jpeg format), which can be used as 'visual title' for the synopsis section of your paper.
6. Please include an author contributions statement after the Acknowledgements section (see <https://www.embopress.org/page/journal/17444292/authorguide>)
7. Please complete the CHECKLIST available at (<https://bit.ly/EMBOPressAuthorChecklist>).

Please note that the Author Checklist will be published alongside the paper as part of the transparent process (<https://www.embopress.org/page/journal/17444292/authorguide#transparentprocess>).

See also figure legend guidelines: <https://www.embopress.org/page/journal/17444292/authorguide#figureformat>

9. Please note that corresponding authors are required to supply an ORCID ID for their name upon submission of a revised manuscript (EMBO Press signed a joint statement to encourage ORCID adoption).

(<https://www.embopress.org/page/journal/17444292/authorguide#editorialprocess>)

Currently, our records indicate that there is no ORCID associated with your account.

Please click the link below to provide an ORCID:

Link Not Available

11. Include a Reagents and Tools Table as part of the Methods section, which can be downloaded from our author guidelines (<https://www.embopress.org/page/journal/17444292/authorguide#structuredmethods>)

*** PLEASE NOTE *** As part of the EMBO Press transparent editorial process initiative (see our Editorial at <https://dx.doi.org/10.1038/msb.2010.72>), Molecular Systems Biology publishes online a Review Process File with each accepted manuscripts. This file will be published in conjunction with your paper and will include the anonymous referee reports, your point-by-point response and all pertinent correspondence relating to the manuscript. If you do NOT want this File to be published, please inform the editorial office at msb@embo.org within 14 days upon receipt of the present letter.

Reviewer #1:

The manuscript from Hong et al describes a multiplexed method for high throughput drug screening of inhibitors of viral proteases. The method involves the use of pooled collections of yeast strains that serve as biosensors of protease activity. Induction of protease expression leads to impaired cell growth, which can be rescued by inhibition. By DNA barcoding each strain, inhibition of activity can be assayed for several proteases concurrently using a NGS DNA sequencing analysis. The authors performed numerous controls to validate the robustness of the approach. They selected 40 viral proteases of interest and performed a multiplexed screen with 2,480 compounds. Several hits were observed with various selectivities, and many hits were validated. Two sets of compounds were selected for follow up including a chromen-2-one set of coronavirus 3CL inhibitors and a pyridinone inhibitor of coronavirus papain-like proteases. Overall, the studies conducted were thorough and clearly show that the approach is effective. This reviewer does not see a need for significant additional experiments. It was nice to see that the authors are making the full data set available, which could be a valuable resource for others. We had only minor concerns about the manuscript and would suggest only some slight changes.

Minor concerns:

1. Data analysis and use of "magratio" as measure of effect size. It would be great to see a distribution of signal for many replicates of positive and negative controls to get a clearer understanding of the assay robustness. The statistical analysis of the data appears to be rigorous and built in normalization appears to be critical for detection inhibition with good fidelity. It would be useful if the authors could include an analysis using the z-prime metric, which is the standard effect size measure for high throughput screening. This may help readers relate the robustness of this approach to other screens. Alternatively, the authors may want to include a further justification for the use of their magratio metric. Also, there are additional factors that may be useful to include in the discussion of their data analysis. Do they observe heteroskedasity in their data variance? Would a log transformation be appropriate? How much sequencing depth per compound per strain is required to get adequate sampling? This would indicate how much multiplexing capacity is possible with this approach and current DNA sequencing capabilities.
2. Choice of yeast as the biosensor. The basic approach for making cell-based biosensors has been published previously by these authors in mammalian cells. It would be nice to include some justification of using yeast. Presumably, this is economically and logistically more feasible for the high-level multiplexing. There are drawbacks with yeast, however. They are notoriously not permeable to many drugs. Perhaps this is why the screening concentration was so high, 25 μ M. It would be great to include some discussion of these concerns and feasibility for other biosensor systems.
3. Covalent inhibitor characterization. Interestingly, both sets of inhibitors selected for follow up appear to be acting covalently. It may be nice to see a time dependent analysis of the IC₅₀ in the biochemical assay follow up. Also, while the nature of the carbamate bond in the PLP inhibitors suggests covalency may be possible, there isn't much data supporting it. We would suggest including time dependent IC₅₀s or mass spec analysis of the labeled protein. ESI analysis would be preferred over the lower resolution MALDI given for 3CL.

Reviewer #2:

This manuscript reported the design of yeast-based screening assay for viral proteases. The assay principle relies on protease-induced cell toxicity, which can be rescued by inhibitor treatment. The authors constructed individual strains with each strain expressing a viral protease. These yeast strains were then pooled in a mixture and used to screen compound libraries. Each strain contains an unique DNA barcode, allowing sequencing of the mixture to determine the growth of each strain. The screening identified both broad-spectrum and target-specific protease inhibitors. Chromen-2-one-containing esters were found to have potent activities against 3CLpro in rescuing yeast growth and inhibiting 3CLpro in the FRET assay. Pyridin-4(1H)-one NSC287495 was identified as a PLpro inhibitor. Follow up lead optimization led to the discovery of several pyridines with potent activities in the yeast growth assay and enzymatic assay. Furthermore, the antiviral activities of several PLpro inhibitors in A549-C3 (ACE2/TMPRSS2) were tested and the EC₅₀ values were from 14.9 to 49.8 μ M. Overall, this assay is innovative and can

speed up the screening of protease inhibitors. Comments are:

1. Did the authors test the antiviral activities of the 3CLpro inhibitors CB6778425, CB6728297, and dCB6762077 in the SARS-Co-2 antiviral assay?
2. The chromen-2-one-containing inhibitors might have fluorescence interference in the FRET-based enzymatic assay. The authors should comment that.
3. For the compound screening, did the authors sequence the mixture at one endpoint or in several time points? In addition, were all the wells sequenced?
4. Although the advantage of the pooled screening is apparent, how about the cost? Is it practical for large scale screening?
5. Since similar indole esters have been reported as SARS-CoV-2 3CLpro inhibitors, the mechanism of action of the chromen-2-one-containing 3CLpro inhibitors is convincing. What about the mechanism of the pyridine compounds in inhibiting PLpro? Are these compounds covalent inhibitors or non-covalent inhibitors? How do they bind to PLpro? Molecular docking should be performed to elucidate their mechanism of action.

Color coding for this document: Reviewer comments in **BLACK**, our responses in **BLUE**, words and edits that are in the manuscript are in **PURPLE**.

Reviewer #1:

The manuscript from Hong et al describes a multiplexed method for high throughput drug screening of inhibitors of viral proteases. The method involves the use of pooled collections of yeast strains that serve as biosensors of protease activity. Induction of protease expression leads to impaired cell growth, which can be rescued by inhibition. By DNA barcoding each strain, inhibition of activity can be assayed for several proteases concurrently using a NGS DNA sequencing analysis. The authors performed numerous controls to validate the robustness of the approach. They selected 40 viral proteases of interest and performed a multiplexed screen with 2,480 compounds. Several hits were observed with various selectivities, and many hits were validated. Two sets of compounds were selected for follow up including a chromen-2-one set of coronavirus 3CL inhibitors and a pyridinone inhibitor of coronavirus papain-like proteases. Overall, the studies conducted were thorough and clearly show that the approach is effective. This reviewer does not see a need for significant additional experiments. It was nice to see that the authors are making the full data set available, which could be a valuable resource for others. We had only minor concerns about the manuscript and would suggest only some slight changes.

Minor concerns:

1. Data analysis and use of "magratio" as measure of effect size. It would be great to see a distribution of signal for many replicates of positive and negative controls to get a clearer understanding of the assay robustness. The statistical analysis of the data appears to be rigorous and built in normalization appears to be critical for detection inhibition with good fidelity. It would be useful if the authors could include an analysis using the z-prime metric, which is the standard effect size measure for high throughput screening. This may help readers relate the robustness of this approach to other screens. Alternatively, the authors may want to include a further justification for the use of their magratio metric. Also, there are additional factors that may be useful to include in the discussion of their data analysis. Do they observe heteroskedasity in their data variance? Would a log transformation be appropriate? How much sequencing depth per compound per strain is required to get adequate sampling? This would indicate how much multiplexing capacity is possible with this approach and current DNA sequencing capabilities.

We appreciate the reviewer's positive comments and thank them for their constructive suggestions. We have investigated the distribution of positive and negative control signals across all plates and present analysis in the below figure. The histograms quantify the proportion of HIV control models in control conditions (DMSO or Lopinavir) as a distribution across all plates. To compute this, we began by normalizing barcoded strain read counts by the total read counts in each well. Then, for each plate, we calculated the mean of well-normalized HIV read counts across all barcodes in a control condition (DMSO or Lopinavir). Repeating this for all plates produced the below histograms.

By comparing HIV in Lopinavir and HIV in DMSO, we computed the z-prime metric (Zhang et al.) as 0.0908 with the positive control mean \pm std of 0.0282 ± 0.0035 and negative control of 0.0067 ± 0.0030 well-normalized read counts. Thus, although Lopinavir properly functioned as a positive control, the assay signal was weak, which necessitated the development of our novel hit calling methods to achieve good discriminative capabilities.

We also investigated the effects of log transformation on our data, and we compared the z-prime metric across different transformations. Where averaging well-normalized read counts across model barcodes yielded a z-prime metric of 0.0908, log-transforming and then normalizing by the well log-counts resulted in a z-prime metric of -0.5234. We further explored whether normalizing by the well read counts is beneficial, and indeed it was. Without well normalization, the z-prime metrics with and without log transformation are -1.4080 and -0.9975, respectively. Therefore, while log transformation improved the z-prime metric when considering raw counts, the most influential preprocessing step was normalizing counts to the total reads per well. Although the z-prime metric may suggest that the assay cannot precisely distinguish viral inhibitors, we were able to develop a robust computational pipeline that leverages our built-in controls to overcome this challenge.

To explore the potential that our data is heteroscedastic, we plotted the coefficient of variation (CV) and the variance of our well-normalized control gene reads across barcodes of every plate. In DMSO control wells, we observed that the variance and CV are unimodal with minor skew, as seen in subfigures (A) and (B), suggesting that our negative controls are homoscedastic. Similarly, the positive control well variance and CV in subfigures (C) and (D) are unimodal, but they exhibit stronger skew than the negative controls. To further explore this skew, we partitioned the positive control well coefficients of variation by thresholding at 0.10 where there is a visual separation in the Lopinavir Coefficient of Variation histogram; from this threshold, we confirmed that this partition did not separate plates nor individual experimenters, so we believe that this distribution is a single skewed one rather than two subpopulations.

Lastly, to estimate the multiplexing capacity of our screen, we explored the question of how much sequencing depth per compound per strain is required to get adequate sampling. To answer this question, we simulated progressively lesser sequencing depth until the magratios of our validated interactions became fully corrupted. Simulating lesser depth was conducted by randomly subsampling well reads, retaining only X% of the original reads for X in [1,2,...,99]. As this simulation was fundamentally stochastic, the number of missed hits fluctuates; nonetheless, we observed a threshold at which our signal was consistently corrupted, as expected. At a sampling depth less than about 20% used in our original screening, our validated magratio signal was lost.

Therefore, based on this simulation, we estimated that we may scale the assay multiplexing by approximately 5 fold without further modification.

In light of the reviewers comments we have added the following text to the manuscript:

LINE 386:

“Our hit-calling methodology, which relies on the redundant barcodes and behavior of the various in-well controls built into our pool consistently identified reproducible hits, with those with the highest magratios showing concordant increases in their rate of validation. Using a magratio cutoff of 0.3, where 1 represents activity of a screening compound on par with the strong positive control, lopinavir, enabled us to achieve a precision of 0.80. Should future groups desire to capture more potential hits, a lower magratio cutoff could be applied, understanding that while this would increase sensitivity, it would also increase the number of false positives. The use of the more classic z-prime metric (Zhang *et al*, 1999) was also explored during our studies, but found to be unsuitable for accurate hit calling (see Appendix Supplementary Note S1 for additional discussion).”

“Appendix Supplementary Note S1

The distribution of positive and negative control signals across all plates screened was investigated and the analysis presented in the below figure. The histograms quantify the proportion of HIV control models in control

conditions (DMSO or Lopinavir) as a distribution across all plates. To compute this, barcoded strain read counts were first normalized by the total read counts in each well. Then, for each plate, the mean of well-normalized HIV read counts was calculated across all barcodes in a control condition (DMSO or Lopinavir). Repeating this for all plates produced the below histograms.

By comparing HIV in Lopinavir and HIV in DMSO, the z-prime metric (Zhang et al., 1999) was computed as 0.0908 with the positive control mean \pm std of 0.0282 ± 0.0035 and negative control of 0.0067 ± 0.0030 well-normalized read counts. Thus, although Lopinavir properly functioned as a positive control, the assay signal was weak, which necessitated the development of our novel hit calling methods to achieve good discriminative capabilities.

The effects of log transformation on our data was also investigated, and the z-prime metric compared across different transformations. Where averaging well-normalized read counts across model barcodes yielded a z-prime metric of 0.0908, log-transforming and then normalizing by the well log-counts resulted in a z-prime metric of -0.5234. Normalizing by the well read counts was further explored and found to be beneficial. Without well normalization, the z-prime metrics with and without log transformation are -1.4080 and -0.9975, respectively. Therefore, while log transformation improved the z-prime metric when considering raw counts, normalizing counts to the total reads per well was determined to be the most influential preprocessing step. Although the z-prime metric may suggest that the assay cannot precisely distinguish viral inhibitors, the robust computational pipeline that leverages our built-in controls was developed to overcome this challenge.

Zhang JH, Chung TD, Oldenburg KR. A simple statistical parameter for use in evaluation and validation of high throughput screening assays. *Journal of biomolecular screening*. 1999 Apr;4(2):67-73. doi: 10.1177/108705719900400206"

Appendix Figure S20. Dataset exhibits homoscedasticity.

The coefficient of variation (CV) and the variance of well-normalized control gene reads (EYFP, HIV) across barcodes of every plate was plotted for DMSO and Lopinavir wells. In DMSO control wells, we observed that the variance and CV are unimodal with minor skew, as seen in subfigures (A) and (B), suggesting that our negative controls are homoscedastic. Similarly, the positive control well variance and CV in subfigures (C) and (D) are unimodal, but they exhibit stronger skew than the negative controls.

Line 447:

“In future efforts, the platform can be enhanced by increasing the size of the screening pool to enable more targets to be examined at once. Indeed, analysis of the multiplexing capacity of our screen, through simulation of reduced sequencing depth, revealed that the assay may be scaled by approximately five-fold without further modification (Appendix Fig. S19).”

Appendix Figure S19. Multiplexing capacity of the screen. Progressively lesser sequencing depth was simulated via random subsampling of well reads, retaining only X% of the original reads for X in [1,2,...,99]. Lost inhibitors, resulting from corrupted magratio signals at the reduced sampling depth, is shown to appear at less than ~20% of reads used within the original screening, suggesting capacity to scale the current screen by approximately five-fold without further modification.

2. Choice of yeast as the biosensor. The basic approach for making cell-based biosensors has been published previously by these authors in mammalian cells. It would be nice to include some justification of using yeast. Presumably, this is economically and logistically more feasible for high-level multiplexing. There are drawbacks with yeast, however. They are notoriously not permeable to many drugs. Perhaps this is why the screening concentration was so high, 25 μ M. It would be great to include some discussion of these concerns and feasibility for other biosensor systems.

We appreciate the reviewer's acknowledgment of our previous work with developing biosensors using mammalian cells. As the reviewer notes, our choice of yeast as a model system stems from its cost-effectiveness, scalability due to their ability to grow in suspension, rapid growth rate, and its ability to be easily engineered such that we can generate hundreds of clonal barcoded lines in short order. These properties as a whole make yeast an excellent model for our high-throughput screening. The reviewer rightly points out that yeast's multiple drug efflux membrane transporters, such as the pleiotropic drug resistance (PDR) network, limit permeability to certain compounds. In this study, we employed a drug-sensitized *pdr1 Δ prd3 Δ snq2 Δ* strain (Piotrowski et al., 2017) which has been well documented to drastically increase small molecule permeability/retention. In future studies, additional genes involved in regulating small molecule efflux such as YRR1 or PDR8 can also be removed from our screening background to help improve small molecule

accumulation within yeast. We are also considering the potential of alternative cellular chassis systems, such as suspension cultures of mammalian cells like CHO or HeLa. These points have been included in the discussion and are listed below for convenience:

LINE 373:

“There are several innovations that contribute to the versatility and robustness of our screening platform including the use of yeast as a tunable cellular biosensor for protease activity, DNA-barcoding to simultaneously track multiple models at once, and redundant barcoding with multiple in-well controls to increase assay specificity and sensitivity. The choice of yeast as a model system is particularly advantageous due to its cost-effectiveness, scalability from its ability to grow in suspension, rapid growth rate, and ease of genetic engineering, allowing for the generation of hundreds of clonal barcoded lines in a short time frame.”

LINE 454:

“While we employed the drug-sensitized *pdr1Δ prd3Δ snq2Δ* strain (Piotrowski *et al*, 2017), which has been well documented to drastically increase small molecule permeability and retention, additional genes involved in regulating small molecule efflux such as *YRR1* or *PDR8* can also be removed to help further improve small molecule accumulation in yeast. The potential of alternative cellular chassis systems, such as suspension cultures of mammalian cells like CHO or HeLa-S3 may also be considered.”

3. Covalent inhibitor characterization. Interestingly, both sets of inhibitors selected for follow up appear to be acting covalently. It may be nice to see a time dependent analysis of the IC₅₀ in the biochemical assay follow up. Also, while the nature of the carbamate bond in the PLP inhibitors suggests covalency may be possible, there isn't much data supporting it. We would suggest including time dependent IC₅₀s or mass spec analysis of the labeled protein. ESI analysis would be preferred over the lower resolution MALDI given for 3CL.

We appreciate the reviewer's suggestion. To address this, time-dependent inhibition analyses were conducted for both the PLP inhibitors and 3CL inhibitors. As for the PLP inhibitors, the NSC287495 analogs, MAVDA-B-116, -190, -201, -217, and -219, exhibited a downward trend in IC₅₀ as a function of pre-incubation time with SARS-CoV-2 PLP as shown below, suggesting a covalent interaction between the enzyme and compounds while the control, GRL0617, a known non-covalent compound, did not.

Moreover, in a follow-up experiment within which the pre-incubated enzyme-compound complex was diluted 100-fold with substrate, the MAVDA-series analogs exhibited a high level of inhibition, suggesting an irreversibility between the compound-enzyme complex. In contrast the control, GRL0617, a known non-covalent inhibitor showed a lower level of inhibition upon dilution.

Separately, the 3CL inhibitors, CB6728297, CB6762077 and CB6778425, also exhibited shifts to lower IC50s upon pre-incubation with SARS-CoV-2 3CL, suggesting a covalent inhibition mechanism, further supporting our already generated mass spectrometry analysis results previously shown in Appendix Fig. S10.

We have included the above analyses in the appendix (Appendix Fig. S15 and S9, respectively) and made the following notes in the manuscript:

LINE 327:

“Similar to the results observed in yeast, all tested compounds were found to inhibit the purified protease (IC_{50} values between 3.5-14.2 μM) with MAVDA-B-219 and B-217 found to be the most potent compounds with IC_{50} values of 3.5 μM and 5.8 μM , respectively. (Fig. 5C). Furthermore, a decrease in IC_{50} values as a function of pre-incubation time with the enzyme suggested a covalent mechanism of inhibition (Appendix Fig. S15A), which was also supported by the observation that the pre-incubated compound-enzyme complexes, when diluted 100-fold and added to substrate, still exhibited high levels of inhibition unlike GRL0617, a known non-covalent inhibitor (Appendix Fig. S15B).”

LINE 281:

“The most potent compound, CB6778425, had IC₅₀ and EC₅₀ values of 62 nM and 782 nM, respectively, while the least potent compound, CB6762077, had IC₅₀ and EC₅₀ values of 1.1 μM and 7.7 μM, respectively. Moreover, a downward shift in the IC₅₀s when any of the three compounds were preincubated with the protease suggested that they might function via covalent inhibition (Appendix Fig. S9). To further interrogate this, we conducted mass spectrometry analysis of the SARS-CoV-2 3CL protease treated with CB6778425 and confirmed a ~95 dalton shift in the protease-inhibitor complex, suggesting a covalent modification of the protease which we hypothesize to occur between the catalytic Cys145 and the 2-furoyl group of CB6778425 (Appendix Fig. S10).”

Reviewer #2:

This manuscript reported the design of yeast-based screening assay for viral proteases. The assay principle relies on protease-induced cell toxicity, which can be rescued by inhibitor treatment. The authors constructed individual strains with each strain expressing a viral protease. These yeast strains were then pooled in a mixture and used to screen compound libraries. Each strain contains an unique DNA barcode, allowing sequencing of the mixture to determine the growth of each strain. The screening identified both broad-spectrum and target-specific protease inhibitors. Chromen-2-one-containing esters were found to have potent activities against 3CLpro in rescuing yeast growth and inhibiting 3CLpro in the FRET assay. Pyridin-4(1H)-one NSC287495 was identified as a PLpro inhibitor. Follow up lead optimization led to the discovery of several pyridines with potent activities in the yeast growth assay and enzymatic assay. Furthermore, the antiviral activities of several PLpro inhibitors in A549-C3 (ACE2/TMPRSS2) were tested and the EC₅₀ values were from 14.9 to 49.8 μM. Overall, this assay is innovative and can speed up the screening of protease inhibitors. Comments are:

1. Did the authors test the antiviral activities of the 3CLpro inhibitors CB6778425, CB6728297, and dCB6762077 in the SARS-Co-2 antiviral assay?

We appreciate the reviewer's positive feedback. Indeed, of the three compounds, CB6778425, being the most potent, was evaluated in a SARS-CoV-2 virus inhibition assay but demonstrated only minimal activity, which we make note of in the manuscript:

LINE 290:

“Initial testing of CB6778425 showed minimal activity within a live virus setting, presumably due to a lability in the reactive 2-furyl ester warhead. Future efforts will be required to determine if this reactivity can be modulated while retaining potent on-target activity.”

As stated in the above passage we believe this outcome may be attributed to the instability of the compound's reactive 2-furyl ester warhead. Considering the shared mechanism of action between CB6778425, CB6728297, and CB6762077, we anticipate similar results for the latter two compounds.

2. The chromen-2-one-containing inhibitors might have fluorescence interference in the FRET-based enzymatic assay. The authors should comment that.

We thank the reviewer for this important comment. As the reviewer points out, the chrome-2-one substructure is commonly seen in coumarin fluorophores including the 7-amino-4-methyl-coumarin (AMC) group in our fluorogenic peptide substrate used in the 3CL protease assay. Fortunately, we did not observe any inhibitor-concentration-dependent change in the initial fluorescence when we fully mixed fluorogenic substrate with the

chrome-2-one-containing inhibitors (data below). Therefore, we were then able to monitor fluorescence change over time to evaluate inhibitor potency in the FRET-based enzymatic assay without interference.

3. For the compound screening, did the authors sequence the mixture at one endpoint or in several time points? In addition, were all the wells sequenced?

The mixture was sequenced at one endpoint of 40 hours after incubation with the screened compound to allow for maximal separation of the strains that experienced rescue and growth from the baseline. All wells within a plate, including a series of negative control (DMSO) wells and positive control (lopinavir) wells, were sequenced and analyzed on a per-plate basis to account for batch effects. Details have been outlined in the methods section.

LINE 543:

“Drug screens were conducted in 96-well deep-well plates (VWR) using 1 mL of inducing media (SC -ura -leu GAL) per well. The yeast starter culture was inoculated at 1:1000, and 2.5 μL of compound or DMSO were added to each well. Compound stocks were prepared at 10 mM concentration resulting in a final screening concentration of 25 μM. The plates were then grown for 40 h at 30 °C with shaking at 1000 rpm.

After growth, the optical density (OD595) of the culture was measured using a 96-well plate reader (Tecan) by taking 100 μL from each well. The remaining culture was then processed for DNA extraction using a modified LiOAc-SDS lysis method as done previously (Resnick et al., 2022).”

Line 582:

“Each screened plate consisted of a set of DMSO wells (negative controls; 14 wells) and compound wells (2 lopinavir positive controls and 80 random library compounds). Each well included the same pool of 220 barcoded strains, including the protease models, eYFP negative control, and kinase models. To identify compounds that rescue yeast growth, barcode read counts were used for the analysis and these were processed on a per-plate basis to account for batch effects.”

4. Although the advantage of the pooled screening is apparent, how about the cost? Is it practical for large scale screening?

As the reviewer notes, pooled screening offers considerable time and effort savings compared to multiple single screens. Our platform further reduces costs through several key features, including the use of *S. cerevisiae* as the model organism, which, beyond its scalability, is highly cost-effective to maintain. Additionally, the implementation of high-throughput DNA sequencing is economical. Below we list the costs of

using sequencing as our output excluding the costs of compound acquisition and outgrowth which are similar between our approach and traditional methods:

Library preparation:

Taq polymerase+buffer: \$0.02/well

dNTPs: \$0.02/well

Forward and Reverse Primers: \$0.01/well

PCR plate: \$0.07/well

PCR plate seal: \$0.01/well

Library sequencing:

Illumina Nextseq 550, 75bp single end sequencing: \$0.80/well

Total sequencing costs per compound screened (including PCR technical replicates): \$1.06/well

As noted from our cost breakdown, the actual sequencing of our libraries is the bulk of our assay costs. In the future these costs can be brought down by taking advantage of the increased availability of sequencing facilities offering partial lanes of sequencing on more cost effective sequencing platforms such as the Illumina Novaseq. In addition, the continued emergence of competitors in the short read sequencing space such as Element Biosciences and Pacific Biosciences are also anticipated to put continued pricing pressure to reduce sequencing costs and further promote the cost-efficiency of our approach.

We have now added a brief note regarding the breakdown in sequencing costs (**Appendix Supplementary Note S2**) and added the following text into the discussion section:

LINE 440:

“Here, using a multiplex cell-based drug screening platform, we demonstrate the possibility of identifying inhibitors to a range of viral proteases as well as human proteins at an accelerated pace relative to conventional approaches. Furthermore, while this approach makes use of next generation sequencing, the added cost per compound is ~\$1/well screened (see Appendix Supplementary Note S2 for details), with the bulk of those costs incurring from the short read sequencing and not the sample preparation. As the costs of sequencing continue to decrease, we anticipate substantial reductions in the cost per sample analyzed.”

Appendix Supplementary Note S2

The costs associated with sequencing, excluding those related to compound acquisition and outgrowth (which are comparable between the current and conventional methods), are outlined below:

Library preparation:

Taq polymerase + buffer: \$0.02/well

dNTPs: \$0.02/well

Forward and Reverse Primers: \$0.01/well

PCR plate: \$0.07/well

PCR plate seal: \$0.01/well

Library sequencing:

Illumina NextSeq 550, 75bp single end sequencing: \$0.80/well

Total sequencing costs per compound screened (including PCR technical replicates): \$1.06/well

Further cost reductions are anticipated due to the increased availability of partial sequencing runs at shared facilities, the development of more cost-effective sequencing platforms (e.g., NovaSeq), and the ongoing emergence of competitors in the short-read sequencing market. These factors are expected to drive down sequencing costs even further, enhancing the overall cost-efficiency of our approach.

5. Since similar indole esters have been reported as SARS-CoV-2 3CLpro inhibitors, the mechanism of action of the chromen-2-one-containing 3CLpro inhibitors is convincing. What about the mechanism of the pyridine compounds in inhibiting PLpro? Are these compounds covalent inhibitors or non-covalent inhibitors? How do they bind to PLpro? Molecular docking should be performed to elucidate their mechanism of action.

We thank the reviewer for encouraging us to perform studies to better characterize the mechanism of our PLpro inhibitors. Based on time-dependent inhibition and the dilution of preformed protease-compound complexes, the data suggest that the MAVDA-B series compounds block the PLpro via a covalent mechanism. We have included **Appendix Figure S15** below which show these data:

Appendix Figure S15. Time-dependent inhibition of SARS-CoV-2 PLP.

(A) SARS-CoV-2 PLP time-dependent dose-response curve (DRC) assay. IC_{50} values were extracted from the DRC for the inhibition of purified SARS-CoV-2 PLP and plotted against the pre-incubation time. Error bars denote mean \pm s.d. of five technical replicates. Nonlinear regression was used to determine the IC_{50} s. **(B)** SARS-CoV-2 PLP X100 assay. Inhibition of purified SARS-CoV-2 PLP was measured after pre-incubation of both compound and protein at a high concentration followed by dilution of 100-fold by the substrate. Error bars denote mean \pm s.e.m. of ten technical replicates.

11th Nov 2024

Manuscript Number: MSB-2024-12417R

Title: A multiplex method for rapidly identifying viral protease inhibitors

Author: Alejandro Chavez

Seo Jung Hong

Samuel Resnick

Sho Iketani

Ji Won Cha

Benjamin Albert

Christopher Fazekas

Ching-Wen Chang

Hengrui Liu

Shlomi Dagan

Michael Abagyan

Pavla Fajtová

Bruce Culbertson

Brooklyn Brace

Eswar Reddem

Farhad Forouhar

J. Fraser Glickman

James Balkovec

Brent Stockwell

Lawrence Shapiro

Anthony O'Donoghue

Yosef Sabo

Joel Freundlich

David Ho

Dear Alex,

Thank you for sending us your revised manuscript. We have now received feedback from the two reviewers who evaluated your study. As you will see below, the reviewers are overall satisfied with the performed revisions. Before we can formally accept the manuscript for publication, we would ask you to address some remaining minor issues listed below:

- the remaining minor concerns of Reviewer #1.

On a more editorial level:

1. Please remove figures from the manuscript file. Figures should only be uploaded as individual, high-resolution Figure files.

2. Funding information: The grant number for the Burroughs Wellcome Fund 1017195.01 is missing from the manuscript file, which needs to be added.

3. Please remove the Authors' contribution section from the manuscript file.

4. Appendix:

- Callout for Appendix Figure S20 is missing and should be added.

- The appendix file should be uploaded in PDF format.

5. Figure legends: Please note that information related to n is missing in the legends of figures 4G-I.

6. Please remove the Reagent and Tools table from the manuscript file and upload it as a separate file(.doc) choosing the file type "Reagent Table". The template(.doc) can be found in our author guidelines:

<https://www.embopress.org/page/journal/17444292/authorguide#structuredmethods>

7. Please provide a specific URL for PRJNA901605 dataset in the data availability statement.

8. Please rename "Disclosure statement and competing interests" to "Disclosure and competing interests statement".

9. Source data files need to be organized with one figure per folder and then uploaded as .zip files. E.g. all the Source data files

for figure 1 need to be saved in a single folder and this needs to be zipped and then uploaded as "SD figure 1.zip" file. For EV and/or appendix figures, ZIP together all source data. SD checklist should be uploaded as 'Related Manuscript File'.

When you resubmit your manuscript, please download our CHECKLIST (<https://bit.ly/EMBOPressAuthorChecklist>) and include the completed form in your submission. *Please note* that the Author Checklist will be published alongside the paper as part of the transparent process (<https://www.embopress.org/page/journal/17444292/authorguide#transparentprocess>)

Click on the link below to submit your revised paper.

Kind regards,
Jingyi

Jingyi Hou, PhD
Scientific Editor
Molecular Systems Biology

If you do choose to resubmit, please click on the link below to submit the revision online before 11th Dec 2024.

IMPORTANT: When you send your revision, we will require the following items:

1. the manuscript text in LaTeX, RTF or MS Word format
2. a letter with a detailed description of the changes made in response to the referees. Please specify clearly the exact places in the text (pages and paragraphs) where each change has been made in response to each specific comment given
3. three to four 'bullet points' highlighting the main findings of your study
4. a short 'blurb' text summarizing in two sentences the study (max. 250 characters)
5. a 'thumbnail image' (550px width and max 400px height, Illustrator, PowerPoint or jpeg format), which can be used as 'visual title' for the synopsis section of your paper.
6. Please include an author contributions statement after the Acknowledgements section (see <https://www.embopress.org/page/journal/17444292/authorguide#manuscriptpreparation>)
7. Please complete the CHECKLIST available at (<https://bit.ly/EMBOPressAuthorChecklist>). Please note that the Author Checklist will be published alongside the paper as part of the transparent process (<https://www.embopress.org/page/journal/17444292/authorguide#transparentprocess>).
8. When assembling figures, please refer to our figure preparation guideline in order to ensure proper formatting and readability in print as well as on screen:
<https://bit.ly/EMBOPressFigurePreparationGuideline>
See also figure legend guidelines: <https://www.embopress.org/page/journal/17444292/authorguide#figureformat>
9. Please note that corresponding authors are required to supply an ORCID ID for their name upon submission of a revised manuscript (EMBO Press signed a joint statement to encourage ORCID adoption). (<https://www.embopress.org/page/journal/17444292/authorguide#editorialprocess>)
Currently, our records indicate that the ORCID for your account is 0000-0001-5626-7140.

Please click the link below to modify this ORCID:
Link Not Available
10. Include a Reagents and Tools Table as part of the Methods section, which can be downloaded from our author guidelines (<https://www.embopress.org/page/journal/17444292/authorguide#structuredmethods>)

As a matter of course, please make sure that you have correctly followed the instructions for authors as given on the submission

website.

*** PLEASE NOTE *** As part of the EMBO Press transparent editorial process initiative (see our Editorial at <https://dx.doi.org/10.1038/msb.2010.72> , Molecular Systems Biology will publish online a Review Process File to accompany accepted manuscripts. When preparing your letter of response, please be aware that in the event of acceptance, your cover letter/point-by-point document will be included as part of this File, which will be available to the scientific community. More information about this initiative is available in our Instructions to Authors. If you have any questions about this initiative, please contact the editorial office (msb@embo.org).

Reviewer #1:

We are satisfied with the modifications made to the manuscript from Hong et al in response to reviewer comments. Overall, the comments were addressed thoroughly and have made the paper stronger. We had just a couple minor concerns regarding data analysis.

- a. We would recommend replacing the histogram shown in Appendix Supplementary Note S1 with a scatter plot of the raw data points, or perhaps also including the scatter plot in addition to the histogram. The crude binning of the histogram makes assessment of the variance challenging visually.
- b. A conclusion in the Appendix Supplementary Note S1 of the log transformation data states that "log transformation improved the z-prime metric". However, the reported value was negative (-0.5234). This was worse than the reported value for the well-normalized read count (0.0908). Negative values indicate a less robust assay. Perhaps this was a typo. We would suggest clarification here.
- c. Significant figures should be kept in mind. The variance of the measurement establishes the number of significant figures. So, for example, the value reported for the positive control mean in Appendix Supplementary Note S1 was 0.0282 +/- 0.0035. This should be reported as 0.028 +/- 0.004. We would suggest changing reported values accordingly throughout. Also, we would suggest reporting the z-prime values then with 2 significant figures.

Reviewer #2:

Comments were properly addressed.

Reviewer #1:

We are satisfied with the modifications made to the manuscript from Hong et al in response to reviewer comments. Overall, the comments were addressed thoroughly and have made the paper stronger. We had just a couple minor concerns regarding data analysis.

a. We would recommend replacing the histogram shown in Appendix Supplementary Note S1 with a scatter plot of the raw data points, or perhaps also including the scatter plot in addition to the histogram. The crude binning of the histogram makes assessment of the variance challenging visually.

We appreciate the reviewer's suggestion and have replaced the figure in Appendix Supplementary Note S1 with the plot shown below:

b. A conclusion in the Appendix Supplementary Note S1 of the log transformation data states that "log transformation improved the z-prime metric". However, the reported value was negative (-0.5234). This was worse than the reported value for the well-normalized read count (0.0908). Negative values indicate a less robust assay. Perhaps this was a typo. We would suggest clarification here.

We thank the reviewer for this important comment. To clarify our intended message, we have revised the section as outlined below:

Appendix Supplementary Note S1

The distribution of positive and negative control signals across all plates screened was investigated and the analysis presented in the below figure. The histograms quantify the proportion of HIV control models in control

conditions (DMSO or Lopinavir) as a distribution across all plates. To compute this, barcoded strain read counts were first normalized by the total read counts in each well. Then, for each plate, the mean of well-normalized HIV read counts was calculated across all barcodes in a control condition (DMSO or Lopinavir). Repeating this for all plates produced the below histograms.

By comparing HIV in Lopinavir and HIV in DMSO, the z-prime metric (Zhang et al., 1999) was computed as 0.091 with the positive control mean \pm std of 0.028 ± 0.004 and negative control of 0.0067 ± 0.0030 well-normalized read counts. Thus, although Lopinavir properly functioned as a positive control, the assay signal was weak, which necessitated the development of our novel hit calling methods to achieve good discriminative capabilities.

We investigated the effects of log transformation on our data and compared the z-prime metric compared across different transformations. Starting with raw read counts, log transformation increases the z-prime metric from -1.4 to -1.0. By contrast, well-normalization increases the z-prime metric from -1.4 to 0.091. Combining these normalizations, however, does not further increase z-prime; log transformation along with well-normalization brings the z-metric down to -0.52. Although the z-prime metric may suggest that the assay cannot precisely distinguish viral inhibitors, the robust computational pipeline that leverages our built-in controls was developed to overcome this challenge.

Zhang JH, Chung TD, Oldenburg KR. A simple statistical parameter for use in evaluation and validation of high throughput screening assays. *Journal of biomolecular screening*. 1999 Apr;4(2):67-73. doi: 10.1177/108705719900400206

c. Significant figures should be kept in mind. The variance of the measurement establishes the number of significant figures. So, for example, the value reported for the positive control mean in Appendix Supplementary Note S1 was 0.0282 +/- 0.0035. This should be reported as 0.028 +/- 0.004. We would suggest changing reported values accordingly throughout. Also, we would suggest reporting the z-prime values then with 2 significant figures.

As per the reviewer's suggestion, the values reported in Appendix Supplementary Note S1 have been revised to consistently display two significant figures throughout, including the z-prime values, as shown in the excerpt above.

Reviewer #2:

Comments were properly addressed.

2nd Dec 2024

Manuscript number: MSB-2024-12417RR

Title: A multiplex method for rapidly identifying viral protease inhibitors

Dear Alex,

Thank you again for sending us your revised manuscript. We are now satisfied with the modifications made and I am pleased to inform you that your paper has been accepted for publication.

Kind regards,
Jingyi

Jingyi Hou, PhD
Scientific Editor
Molecular Systems Biology
